# KV Cache Steering for Controlling Frozen LLMs

## Abstract

We propose *cache steering*, a lightweight method for implicit steering of language models via a one-shot intervention applied directly to the key-value cache. To validate its effectiveness, we apply cache steering to induce chain-of-thought reasoning in small language models. Our approach constructs steering vectors from reasoning traces, obtained either from teacher models (e.g., GPT-4o) or existing human annotations, that shift model behavior toward more explicit, multi-step reasoning without fine-tuning or prompt modifications. Experimental evaluations on diverse reasoning benchmarks demonstrate that cache steering improves both the qualitative structure of model reasoning and quantitative task performance. Additional experiments show that the method also scales to larger models and yields further gains on challenging datasets such as GPQA and MATH. Compared to prior activation steering techniques that require continuous interventions, our one-shot cache steering offers substantial advantages in terms of inference latency, hyperparameter stability, and ease of integration with existing inference APIs. Beyond mere reasoning induction, we show that cache steering enables controllable transfer of reasoning styles (e.g., stepwise, causal, analogical), making it a practical tool for behavior-level guidance of language models.

## 1 Introduction

The ability of large language models to perform complex reasoning is a key driver of their increasing utility. However, this potential is not always spontaneously realized, especially in smaller models which may possess latent reasoning capabilities that require specific guidance to activate. Traditional methods for uncovering these abilities, such as supervised fine-tuning or few-shot prompting with chain-of-thought examples, can be effective but often demand significant data or intricate prompt design. The question then arises: can we develop more lightweight interventions to unlock and steer these inherent reasoning processes post-training?

One promising direction is activation steering (Turner et al., 2024; Rimsky et al., 2024), which aims to guide model behavior by directly modifying its internal hidden states. While promising for its ability to influence outputs without retraining, activation steering often requires continuous interventions at each token generation step throughout the decoding process to be effective (Wehner et al., 2025). This continuous manipulation can introduce instability, making the outcomes highly sensitive to hyperparameter choices (e.g., targeted layers, intervention strength) and potentially leading to a degradation in generation quality.

To address these issues, we introduce a method called **cache steering**. Our approach operates by making a targeted, one-time modification directly to the key-value cache of a Transformer model, typically after the cache has been populated by an initial prompt. By applying steering vectors, derived either from reasoning traces generated by a capable teacher model like GPT-4o or from existing human/dataset annotations, to these cached key and value representations, we can guide the reasoning trajectory of language models. This single intervention, applied before token generation begins, effectively steers the model towards more explicit, multi-step reasoning without altering model weights or requiring complex prompt modifications. Compared to activation steering, which applies interventions at every decoding step, cache steering avoids cascading effects, is robust to hyperparameter choices, introduces virtually no runtime cost, and seamlessly integrates with standard inference pipelines.

Figure 1: **Activation steering vs. Cache steering**. Activation steering (left) injects vectors into hidden states *dynamically* during decoding, typically at a single chosen layer. Intervening at multiple layers is possible but often amplifies effects across the network, making the method sensitive to tuning and prone to instability. Cache steering (right) instead modifies the pre-computed KV cache once after the prefilling step. Because these cached representations are *fixed*, the intervention can be applied consistently across all layers and then implicitly influences future tokens, leading to stable and efficient inference without repeated runtime injections.

We demonstrate that our method improves reasoning structure and, in many cases, task accuracy on multiple benchmarks, including GSM8K, ARC-Challenge, CSQA, and PIQA, and further scales to larger models and harder datasets such as GPQA and MATH. Beyond simple reasoning induction, cache steering enables *controllable transfer of reasoning styles* (e.g., stepwise, causal, analogical), illustrating its value as a practical tool for behavior-level guidance.

Overall, our key contributions are as follows:

- We propose **cache steering**, a one-shot modification of the KV cache that provides a lightweight and production-ready alternative to activation steering and fine-tuning.
- We demonstrate that cache steering can **distill reasoning styles** from teacher models or existing reasoning traces into smaller models without weight updates or prompt augmentation.
- We conduct extensive evaluations across multiple model families and benchmarks, analyze efficiency and stability, and provide additional results on larger models and challenging reasoning datasets.

## 2 RELATED WORK

**Reasoning and Chain-of-Thought prompting.**    A widely adopted approach to enhance reasoning abilities in LLMs involves demonstrating example solutions to the problem (in-context learning or ICL) that contain a step-by-step reasoning process (Chain-of-Thought or CoT) in a prompt to the language model (Brown et al., 2020; Wei et al., 2022), a technique known as few-shot prompting. Zero-shot variants of CoT prompting simplify this approach by adding instructions such as "Let's think step by step" to elicit step-by-step reasoning without the need for example demonstrations (Kojima et al., 2022).

Recent work shows that reinforcement learning can lead to remarkable reasoning capabilities, which can be effectively distilled into smaller models through supervised fine-tuning (Guo et al., 2025). These findings suggest that it is not enough to just trigger CoT reasoning in the language models, but the **style** of the reasoning matters. This motivates our approach, which aims to directly steer small models toward reasoning behavior reminiscent of larger teacher models via cache-level interventions.

**Activation steering.**    Activation steering, also known as representation engineering, is a technique used to control the generation process of LLMs implicitly by manipulating their intermediate activations during decoding, typically through linear interventions (Rimsky et al., 2024; Turner et al., 2024). Multiple works have applied activation steering to induce or suppress specific behaviors in models without retraining. The examples include sentiment, topic and style control (Turner et al., 2024); function steering (Todd et al., 2024; Postmus & Abreu, 2024), removing or inducing refusal behavior, (Lee et al., 2025), toxicity reduction (Turner et al., 2024), truthfulness (Wang et al., 2025),

editing factual knowledge (Yin et al., 2024), reasoning induction (Zhang & Viteri, 2025; Galichin et al., 2025), reasoning compression (Azizi et al., 2025; Chen et al., 2025) and other (Wehner et al., 2025).

In its most basic form, activation steering involves two steps: vector extraction and injection of the vector into the activations of the model at inference. The vector extraction stage involves computing a "steering vector", which is commonly done by aggregating activations from pairs of positive prompts with desired behavior and negative or sometimes neutral prompts, forming a contrastive set $C = \{(p_0^+, p_0^-), (p_1^+, p_1^-), ..., (p_N^+, p_N^-)\}$. The most common aggregation method is Difference-in-Means (Wehner et al., 2025), which is identical to Mean-of-Differences when the vectors are paired:

$$\boldsymbol{s}_l = \frac{1}{N} \sum_{(p^+, p^-) \in C} f_l(p^+) - f_l(p^-)$$

where $f_l$ represents the part of the Transformer model (e.g., the whole decoder layer) at layer $l$ and $N$ is the number of examples in the contrastive dataset $C$. To steer the model's output, the steering vector is added to the activations of specific layers during inference:

$$\boldsymbol{h}_l^* = \boldsymbol{h}_l + c\boldsymbol{s}_l$$

where $\boldsymbol{h}_l$ represents the activations at layer $l$ before steering, $\boldsymbol{s}_l$ is a steering vector extracted from layer $l$, and $c$ is a coefficient that determines the strength of the steering.

It is important to mention that the vector can be extracted and applied to different token positions, layers, and parts of the model, which are treated as hyperparameters or design choices. Usually, it is a common practice to perform a grid search to determine the layers to apply steering to and the value of the steering strength coefficient $c$ (Turner et al., 2024; Lee et al., 2025; Wang et al., 2025; Dong et al., 2024; Rimsky et al., 2024; Wang et al., 2024; Stolfo et al., 2025; Zhang & Viteri, 2025; Postmus & Abreu, 2024).

While activation steering offers a tool for model control, it typically requires continuous intervention during generation (Wehner et al., 2025), which can be expensive and can lead to unstable generations. Several studies address this hyperparameter sensitivity through dynamic steering, where the steering strength is adapted throughout decoding. Methods such as Dynamic Activation Composition (Scalena et al., 2024) and Episodic-Memory Steering (Tran et al., 2025) dynamically adjust the steering strength coefficients using KL-guided updates or memory-bank interpolation to mitigate oversteering. However, these approaches require continuous modification of activations, additional forward passes, or auxiliary data structures, making them not practical for real-world scenarios.

Our work takes a different approach: rather than adapting activation edits on the fly, we shift the intervention target entirely to the static key–value (KV) cache, allowing a single post-prefill modification that does not propagate vertically through the network. This removes the need for dynamic schedules altogether and enables a one-shot intervention that is both efficient and stable at inference time. We discuss the steering amlification effect in detail in Section 3.4.

**Cache manipulation.** Another emerging line of research explores the idea of modifying the key-value (KV) cache from the memory and efficiency perspective (Li et al., 2024; Liu et al., 2025a; Ge et al., 2024; Mu et al., 2023). These approaches aim to reduce the memory footprint or compress contextual representations through KV cache manipulation. Building on this idea, Liu et al. (2025b) introduced a method for augmenting the KV cache to improve the performance on tasks that require reasoning abilities. The authors use a differentiable "coprocessor", which allows augmenting the KV cache as a pre-generation step instead of modifying activations directly during the forward pass. However, in order to augment the KV cache, the method requires training a separate model, which makes this method less practical than pure activation steering methods introduced in the previous subsection. In contrast, our approach aims to use the KV cache as a target for behavioral control in small models without training auxiliary modules.

## 3 CACHE STEERING

We introduce **cache steering**, a lightweight method for inducing structured reasoning in language models by applying steering vectors directly to the key-value cache. Unlike traditional activation steering methods, which modify intermediate hidden states during generation, our approach modifies the cached keys and values associated with specific tokens, enabling a one-shot intervention that can be precomputed and reused. This technique is compatible with standard inference APIs and does not require model fine-tuning or prompt engineering.

## 3.1 PRELIMINARIES

Transformer-based language models rely on the self-attention mechanism, which operates on sets of query, key, and value vectors to compute contextualized token representations. For a given input sequence, the attention output at layer $l$ is computed as:

$$\text{Attention}(\boldsymbol{Q}^l, \boldsymbol{K}^l, \boldsymbol{V}^l) = \text{softmax}\left(\frac{\boldsymbol{Q}^l(\boldsymbol{K}^l)^\top}{\sqrt{D_h}}\right)\boldsymbol{V}^l$$

where $\boldsymbol{Q}^l, \boldsymbol{K}^l, \boldsymbol{V}^l \in \mathbb{R}^{T \times H \times D_h}$ are the query, key, and value tensors at layer $l$, $T$ is the sequence length, $H$ is the number of attention heads, and $D_h$ is the dimensionality of each head.

During autoregressive decoding, the model stores the keys $\boldsymbol{K}^l$ and values $\boldsymbol{V}^l$ corresponding to previously processed tokens, which is known as a key-value (KV) cache. These cached tensors are used to efficiently compute attention for each new token without recomputing representations for the entire sequence. Importantly, these cache entries can be precomputed and reused across multiple examples (such as caching a system prompt), which is especially useful in scenarios involving large models or repeated inference over similar inputs. This makes the KV cache a potential target for behavioral interventions, offering compatibility with real-world settings.

## 3.2 EXTRACTING KEY-VALUE STEERING VECTORS

Similarly to activation steering, we construct a contrastive set of prompt pairs $C = \{(p_0^+, p_0^-), (p_1^+, p_1^-), ..., (p_N^+, p_N^-)\}$ to extract the key-value steering vectors. We refer to prompts that demonstrate the desired behavior as positive and the prompts without such behavior as negative. We discuss the details of how the positive and negative prompts are constructed for the reasoning induction task in Section 3.5.

For each contrastive pair of examples, we make a forward pass and extract the keys and values vectors from the designated token position (typically the final token of the input prompt). The vectors are then aggregated using the Mean-of-Differences method:

$$\boldsymbol{S}_l^k = \frac{1}{N} \sum_{(p^+, p^-) \in C} f_l^k(p^+) - f_l^k(p^-) \qquad \boldsymbol{S}_l^v = \frac{1}{N} \sum_{(p^+, p^-) \in C} f_l^v(p^+) - f_l^v(p^-)$$

where $f_l$ is a Transformer layer, $\boldsymbol{S}_l^k \in \mathbb{R}^{H \times D_h}$ and $\boldsymbol{S}_l^v \in \mathbb{R}^{H \times D_h}$ are the resulting steering tensors at layer $l$, with $H$ denoting the number of attention heads and $D_h$ their dimension. By taking the difference between positive and negative examples and averaging across multiple contrastive pairs, we aim to isolate a directional signal associated with target behavior while minimizing the amount of noise introduced by information from individual examples.

## 3.3 APPLYING KEY-VALUE STEERING VECTORS

At inference time, we perform a standard forward pass on the input prompt to populate the KV cache. Then, at each layer $l$, we modify the cached key and value vectors at a target token position of the KV cache as follows:

$$\boldsymbol{V}_l^* = \boldsymbol{V}_l + c^v \boldsymbol{S}_l^v \qquad \boldsymbol{K}_l^* = \boldsymbol{K}_l + c^k \boldsymbol{S}_l^k$$

where $\boldsymbol{K}_l, \boldsymbol{V}_l \in \mathbb{R}^{H \times D_h}$ are the original cached key and value vectors at layer $l$, and $\boldsymbol{S}_l^k, \boldsymbol{S}_l^v \in \mathbb{R}^{H \times D_h}$ are the steering vectors, and $c^k, c^v \in \mathbb{R}$ are scalar coefficients controlling the steering strength. Then the generation proceeds as usual using the modified cache.

## 3.4 ELIMINATING STEERING AMPLIFICATION

Cache steering differs fundamentally from traditional activation steering in *how* and *when* interventions are applied. Activation steering can amplify across layers and timesteps, making it unstable and sensitive to hyperparameters. Cache steering eliminates this amplification by modifying the fixed KV cache once after prefilling, as illustrated in Figure 1. Below, we outline the core intuition behind this contrast.

At a specific timestep $t$, activation steering explicitly affects the current hidden state at a chosen layer $l$. This modification propagates both *vertically* through all subsequent layers $l+1$ to $l+N$ and

*horizontally* into future tokens as decoding continues. Because interventions accumulate across layers and timesteps, small changes can compound into "oversteering," which can negatively affect generation quality. This makes the method highly sensitive to hyperparameters such as steering strength and application layer (Turner et al., 2024; Lee et al., 2025; Wang et al., 2025; Dong et al., 2024; Rimsky et al., 2024; Wang et al., 2024; Stolfo et al., 2025; Zhang & Viteri, 2025; Postmus & Abreu, 2024).

In contrast, cache steering modifies the *fixed* key and value representations of *past* tokens after the prefilling stage. These cached representations are no longer transformed through the network and can therefore be adjusted vertically across all layers without risk of compounding. Future tokens then attend to this modified cache, so the steering effect propagates horizontally across the tokens during decoding. This one-shot intervention avoids cascading effects, allowing cache steering to be both stable to hyperparameters and efficient at runtime.

In short, cache steering replaces the compounding per-step interventions of activation steering with a single post-prefill modification that avoids amplification, yielding a stable and efficient mechanism for guiding model behavior.

### 3.5 IMPLEMENTATION DETAILS

**Contrastive set construction.** To extract steering vectors, we construct a contrastive dataset consisting of paired prompts. Each pair includes a **positive example** (containing explicit chain-of-thought reasoning) and a **negative example** (containing only the final answer). Each contrastive prompt is created using few-shot in-context learning (ICL) examples. Specifically, both the positive and negative prompts include $n$ ICL examples followed by a question and a generation prompt. The positive and negative prompts differ only in the presence of reasoning steps in the ICL examples (see Appendix C.1 for more details and an illustrative example).

**Extraction and application positions.** We extract key and value vectors from the final token of the prompt, which typically corresponds to the last token of the generation prompt depending on the model's chat template (e.g. `"\n\n"` in `"assistant\n\n"`). During inference, we aim to apply cache steering to the same logical position in the prompt as used during extraction. However, due to the autoregressive decoding mechanism (see Section 3.1), the KV cache is populated only after each token is processed. To ensure alignment, we append a neutral offset token (e.g., a newline or whitespace) to the prompt, so that the KV representation of the final token can be used in the generation of the next tokens. This ensures the cache steering affects the intended location. Details on token alignment and cache offset are provided in Appendix C.5.

**Hyperparameters.** As with activation steering, the steering strength coefficients of key and value vectors are treated as hyperparameters. Since we are interested in distilling reasoning behaviors from larger models, we additionally treat the number of contrastive pairs and the number of in-context examples in each pair as additional hyperparameters. Similarly to other steering approaches, we perform a small grid search over the hyperparameters to obtain reasonable values for each model-dataset pair (Turner et al., 2024; Lee et al., 2025; Wang et al., 2025; Dong et al., 2024; Rimsky et al., 2024; Wang et al., 2024; Stolfo et al., 2025; Zhang & Viteri, 2025; Postmus & Abreu, 2024). We find that steering coefficients tend to lie within consistent ranges across tasks, suggesting robustness in the method's behavior. More on this in Section 5.3. The full list of hyperparameters can be found in Appendix G.

## 4 EXPERIMENTAL SETUP

**Datasets.** We use four common reasoning benchmarks for the evaluation: GSM8K (Cobbe et al., 2021), CommonsenseQA (Talmor et al., 2018), ARC-Challenge (Clark et al., 2018), and PIQA (Bisk et al., 2020). These datasets span arithmetic reasoning, commonsense inference, scientific questions, and physical commonsense reasoning. For each dataset, we generate elaborate step-by-step answers to a subset of questions from the corresponding training sets using GPT-4o, which are then used in positive examples in the contrastive set. The details on the specific prompt used to generate these steps and the generation procedure can be found in the Appendix C.6. Steering vectors are computed using the training set, while evaluation is performed on the corresponding test sets.

Table 1: **Comparison of baselines, activation steering, and cache steering on four reasoning benchmarks**. We evaluate six models of different sizes on GSM8K, ARC-Challenge, CSQA, and PIQA using both greedy decoding (left block) and sampling-based decoding (right block). Results show that cache steering consistently improves reasoning performance, often outperforming both baseline and activation steering. Combining cache steering with CoT prompting yields further gains in more than half of the cases. Numbers in parentheses denote standard deviation across 5 sampled generations per input; the sampling block highlights that cache steering produces a *stable shift in logits*, as reflected by consistently better or on-par performance under stochastic decoding.

| Dataset | Model | Greedy | | | | | Sampling | |
|---|---|---|---|---|---|---|---|---|
| | | Baseline | CoT prompt | Activation steering | Cache steering | Cache steering + CoT prompt | Baseline | Cache steering |
| ARC-c | SmolLM2-360M | 24.32 | 26.62 | 24.06 | **27.13** | 25.26 | 24.16 (1.13) | **24.52** (0.87) |
| | Llama-3.2-1B | 53.67 | 53.75 | 53.84 | 55.03 | **56.14** | 52.29 (0.81) | **53.16** (1.44) |
| | Llama-3.2-3B | 74.32 | 77.13 | 74.23 | 79.27 | **79.52** | 74.64 (0.36) | **77.71** (0.82) |
| | Qwen2-0.5B | 39.51 | 37.20 | **40.69** | 40.36 | 38.82 | **38.05** (0.21) | 35.96 (0.93) |
| | Llama-3.1-8B | 83.11 | 84.98 | 84.64 | **85.58** | 85.24 | 82.66 (0.28) | **85.09** (0.64) |
| | Phi-4-mini | 84.56 | 86.69 | 86.18 | **87.97** | 86.77 | 83.46 (0.56) | **87.2** (0.62) |
| GSM8K | SmolLM2-360M | 8.49 | **10.39** | 7.66 | 8.95 | **10.39** | **8.08** (0.38) | 7.87 (0.4) |
| | Llama-3.2-1B | 45.56 | 46.10 | 45.41 | 46.32 | **47.16** | 43.71 (0.83) | **43.88** (1.22) |
| | Llama-3.2-3B | 68.54 | 71.57 | 68.38 | 67.17 | **72.10** | **68.22** (0.43) | 67.57 (1.22) |
| | Qwen2-0.5B | 17.44 | 24.94 | 23.81 | 18.04 | **25.47** | **16.94** (1.08) | 16.48 (0.4) |
| | Llama-3.1-8B | 76.34 | 77.56 | 76.50 | 75.81 | **77.86** | **75.94** (0.62) | 75.22 (0.58) |
| | Phi-4-mini | 77.94 | 74.68 | 75.89 | **78.47** | 75.74 | **77.48** (0.62) | 77.1 (0.66) |
| CSQA | SmolLM2-360M | 19.74 | 22.11 | 19.66 | 21.95 | **22.52** | 20.02 (1.77) | **21.31** (0.67) |
| | Llama-3.2-1B | 53.56 | 54.71 | 54.14 | **55.20** | 53.56 | **51.45** (0.64) | 50.78 (0.73) |
| | Llama-3.2-3B | 70.27 | **72.56** | 69.12 | 72.32 | 72.48 | 70.09 (0.78) | **70.40** (1.1) |
| | Qwen2-0.5B | **47.42** | 44.31 | 45.95 | 46.03 | 45.37 | **45.67** (1.18) | 42.36 (1.11) |
| | Llama-3.1-8B | 73.87 | 74.04 | 73.30 | **75.27** | 74.37 | 73.92 (0.37) | **74.27** (1.01) |
| | Phi-4-mini | 69.78 | 70.11 | 69.29 | 70.00 | **70.52** | **68.22** (0.62) | 67.52 (1.08) |
| PIQA | SmolLM2-360M | 50.38 | **52.61** | 49.62 | 51.31 | 52.50 | 48.12 (0.66) | **50.72** (1.03) |
| | Llama-3.2-1B | 65.29 | 64.96 | 61.48 | 63.76 | **66.43** | **65.02** (0.88) | 62.48 (1.57) |
| | Llama-3.2-3B | 69.42 | **76.93** | 72.31 | 73.34 | 76.88 | 68.35 (0.28) | **71.83** (0.66) |
| | Qwen2-0.5B | 52.12 | 53.43 | 53.43 | 54.57 | **55.55** | 51.82 (0.68) | **53.86** (0.44) |
| | Llama-3.1-8B | 80.03 | 81.61 | 80.25 | 82.86 | **83.13** | 79.08 (0.39) | **83.41** (0.52) |
| | Phi-4-mini | 78.29 | 79.59 | **80.74** | 79.33 | 80.25 | 77.19 (0.63) | **79.46** (0.68) |

**Models.** We evaluate cache steering on small instruction-tuned models from four families: Llama-3.2 (1B and 3B variants), SmolLM2 (360M), Qwen2 (0.5B), and Phi-4-mini (3.8B) (Grattafiori et al., 2024; Team, 2024; Allal et al., 2025; Abouelenin et al., 2025). Additionally, we add the Llama-3.1 (8B) model to evaluate how cache steering scales beyond the smallest models. The list with the full model names and URLs can be found in Appendix C.8.

**Decoding strategies.** Since cache steering affects internal representations, which result in a shift in output logits, we evaluate our approach using both deterministic and stochastic decoding. For sampling-based decoding, we assess the consistency of steering effects by generating the response with 5 different seeds and comparing that to the baseline generations using the same setup. The generation arguments can be found in Appendix C.4.

**Answer extraction and metrics.** Answer correctness is determined using task-specific heuristics. For GSM8K, we extract the final number mentioned in the output using digit pattern matching (Wang et al., 2023; Wang & Zhou, 2024). For multiple-choice tasks (ARC, PIQA, CSQA), we develop a 4-stage extraction pipeline that uses soft string matching against known answer choices, with failover to constrained decoding. More details on the answer extraction process can be found in Appendix C.2.

**Comparison to activation steering.** In several experiments, we compare cache steering to activation steering. More specifically, we use the CAA method (Rimsky et al., 2024), which is one of the most popular methods for activation steering. In all experiments, we make the best effort to provide a fair

Table 2: **Cache steering consistently increases the length of generated outputs across tasks**. We report the average number of generated tokens under three conditions: baseline decoding, CoT prompting, and cache steering. Results are shown for multiple model sizes averaged across reasoning benchmarks. Cache steering leads to significantly longer outputs, exceeding even CoT-prompted completions, suggesting that the intervention encourages more elaborate reasoning, even without explicit prompting.

| Model | Baseline | CoT | Cache Steering |
|---|---|---|---|
| SmolLM2-360M | 73.5 (94.1) | 194.8 (27.7) | **294.2** (52.4) |
| Qwen2-0.5B | 100.0 (59.5) | 167.8 (52.1) | **225.0** (50.6) |
| Llama-3.1-8B | 156.0 (36.7) | 174.8 (18.6) | **297.2** (61.0) |
| Llama-3.2-1B | 121.8 (50.7) | 161.8 (27.5) | **291.2** (122.4) |
| Llama-3.2-3B | 160.2 (37.7) | 181.0 (29.6) | **284.5** (95.8) |
| Phi-4-mini | 107.8 (33.5) | 211.0 (20.7) | **328.8** (132.7) |

Table 3: **Results on larger model and harder benchmarks.** Evaluation of cache steering on Llama-3.1-70B-Instruct across ARC-Challenge, GPQA Diamond, and a subset of MATH.

| Dataset | Model | Baseline | CoT prompt | Cache Steering | Cache steering + CoT prompt |
|---|---|---|---|---|---|
| ARC-c | Llama-3.1-70B | 93.00 | 92.91 | **93.52** | 93.17 |
| GPQA Diamond | Llama-3.1-70B | 40.40 | 44.95 | 44.95 | **47.98** |
| MATH (subset) | Llama-3.1-70B | 66.22 | 62.68 | **73.63** | 64.95 |

comparison. The details of how activation steering vectors are extracted and applied can be found in Appendix C.3.

## 5 EXPERIMENTS

### 5.1 INDUCING REASONING VIA CACHE STEERING

To evaluate the effectiveness of cache steering in inducing reasoning behavior, we compare it against several baselines: standard greedy decoding (no intervention), CoT prompting (appending "Let's think step by step" to the prompt), and activation steering. We also evaluate a hybrid approach that combines CoT prompting with cache steering. As shown in Table 1 (greedy part), cache steering consistently outperforms the baseline and often leads to performance gains over the CoT prompting. Furthermore, the combination of CoT prompting with cache steering leads to additional gains in more than half of the cases, indicating the complementary nature of both techniques. Notably, cache steering surpasses activation steering in almost all cases.

Additionally, we report the mean number of generated tokens per model, averaged over all datasets, in Table 2 (results for each dataset-pair can be found in Appendix F). Cache steering leads to longer outputs, suggesting that the intervention encourages reasoning even without explicit prompting. Taking into account the results from both tables, we can conclude that cache steering leads to well-structured reasoning traces, which can be further confirmed by examining qualitative examples in Appendix B.

To complement our main results on small and medium-sized models, we evaluate cache steering on a larger model (Llama-3.1-70B) and more challenging benchmarks. The results in Table 3 show +4.6% accuracy on GPQA Diamond and +7.4% on a MATH subset. These gains are even stronger than those observed on small models, where limited base capabilities can bottleneck the benefits of induced reasoning. This suggests that cache steering has the potential to scale effectively with model size.

**Stability under sampling.** The right side of Table 1 reports results under sampling-based decoding, comparing cache steering to the baseline across multiple models and tasks. We observe that cache steering produces consistent performance improvements or maintains parity with the baseline, indicating that the intervention leads to stable and meaningful *shift in logits*. Rather than injecting noise or introducing erratic behavior, cache steering systematically biases the model toward more

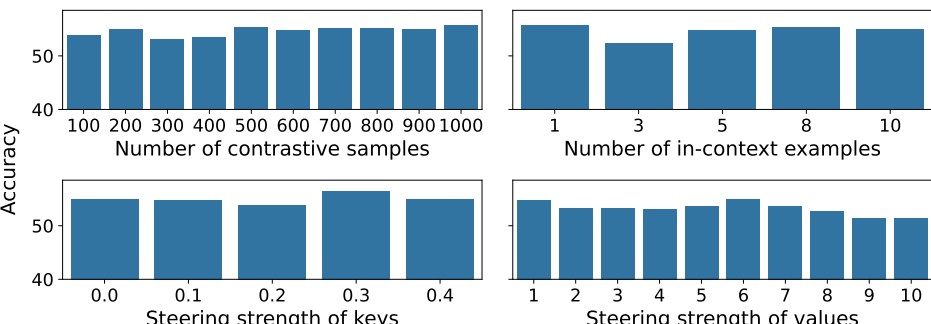

Figure 2: **Cache steering ablations on ARC-c (Llama-3.2-1B-Instruct)**. Accuracy remains stable across contrastive set sizes and key/value steering strengths, with optimal performance around $c^k = 0.3$ and $c^v = 6$. Fewer in-context examples (e.g., 1-shot) yield better steering, likely due to reduced noise. Overall, the method is robust to a range of hyperparameters.

structured reasoning even under stochastic generation. The relatively low standard deviations across runs further support the robustness of the effect.

## 5.2 COMPUTATIONAL OVERHEAD

Cache steering involves only a one-time cache modification after the prefilling step and does not require any additional forward passes compared to the baseline. In contrast, activation steering typically requires continuous interventions at every decoding step for the steering to be effective (Wehner et al., 2025). As shown in Figure 3, cache steering achieves latency nearly identical to the baseline ($\sim$10 ms/token at batch size 1), while activation steering incurs substantially higher time per token ($\sim$15 ms/token, with the gap widening at larger batch sizes). These findings underscore the practical efficiency of cache steering, making it well-suited for real-world deployment scenarios. Full experimental details are provided in Appendix C.7.

## 5.3 ABLATION STUDIES

We conduct ablation experiments on Llama-3.2-1B-Instruct and the ARC-c dataset to assess the sensitivity of cache steering to the primary hyperparameters: 1) number of contrastive pairs used to extract steering vectors, 2) number of few-shot examples per contrastive example, and 3) steering strength coefficients $c^k$ and $c^v$. The results for all the ablation studies can be found in Figure 2.

**Vector extraction.** We vary the number of contrastive pairs from 100 to 1000. The accuracy remains relatively stable across this range, with only minor fluctuations (from 53.1% to 55.7%). This suggests that even small contrastive sets can yield effective steering vectors, though performance tends to improve slightly with larger sets. We also vary the number of ICL examples per prompt from 1 to 10. Interestingly, the best result is achieved with a single example (55.8%), and performance dips slightly at 3-shot (52.4%) before recovering. This non-monotonic trend suggests that reasoning signals may be sensitive to specific examples in the training data.

**Vector application.** More importantly, we observe that cache steering is robust to steering strength variation. Varying the key coefficient $c^k$ between 0.0 and 0.4 results in only minor changes, with the best performance at $c^k = 0.3$ (56.4%). Varying the value coefficient $c^v$ from 1 to 10 shows a peak around $c^v = 6$ (55.0%) and a gradual decline afterward, with performance dropping below 52% beyond $c^v = 8$. Although extreme hyperparameters can lead to slight performance drops, cache steering remains stable to local changes to the coefficients. In contrast, activation steering often exhibits high sensitivity, with small shifts in coefficient values leading to catastrophic generation failures (Rimsky et al., 2024; Turner et al., 2024; Da Silva et al., 2025). We show the sensitivity of activation steering to hyperparameters on a smaller subset of ARC-c in Appendix E.

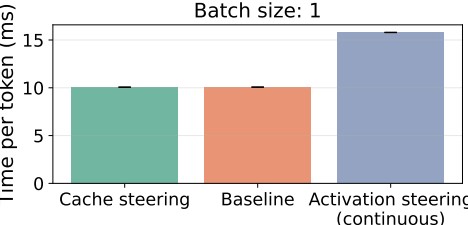 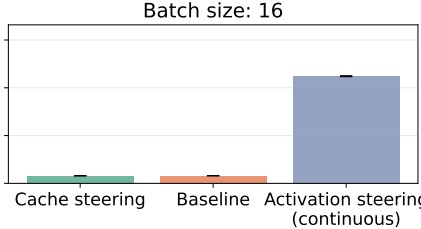

Figure 3: **Cache steering introduces negligible overhead compared to activation steering.** We report average time per token (in milliseconds) for cache steering, activation steering, and the baseline (no intervention) on a single H100 GPU, using batch sizes of 1 and 16. At batch size 1, both the baseline and cache steering run at ∼10 ms/token, while activation steering is slower at ∼15 ms/token; the gap widens further at larger batch sizes. Unlike activation steering, which requires continuous intervention, cache steering adds virtually no runtime cost over baseline inference.

Table 4: **Percentage of generated outputs that exhibit the intended structure when steered using a style-specific vector**. Results demonstrate that cache steering can reliably induce distinct reasoning styles, although its effectiveness varies across styles.

| Metric (%) | Stepwise Reasoning | Strategy+ Execution | Causal Chain | Annotated Deduction | Analogical Reasoning |
|---|---|---|---|---|---|
| Matching Style | **95** | 35 | **95** | 15 | 90 |

> **SmolLM-360M-Instruct on ARC-c**
>
> **Prompt:** *An astronomer observes that a planet rotates faster after a meteorite impact. Which is the most likely effect of this increase in rotation?*
>
> **Analogical Reasoning:** Just like Earth, planets with higher rotation rates tend to have more massive cores. When...
>
> **Casual Chain:** If a planet rotates faster after a meteorite impact, it is likely that the planet's rotation rate...
>
> **Annotated deduction:** [Faster rotation] [Planetary density will decrease] [Planetary years will become...]
>
> **Stepwise reasoning:** Step 1: Understand the impact of the meteorite on the planet's rotation. Step 2: Identify...
>
> **Strategy execution:** Strategy: To determine the most likely effect of a meteorite impact on a planet's rotation...

Figure 4: **Example outputs on a single ARC-c question, using different style-specific vectors**. Each generation reflects the structure of the steering traces used to construct the corresponding vector.

## 5.4 STYLE TRANSFER

To explore whether cache steering can be used to distill distinct reasoning styles from a teacher model, we evaluate how the stylistic form of the reasoning traces used to extract the vectors affects the response structure. For this experiment, we construct reasoning traces of 5 different styles (definitions and experiment details in Appendix A) for a subset of ARC-Challenge questions and extract one steering vector per style. Table 4 reports the percentage of generated responses that match the intended structure for each style-specific steering vector using the SmolLM-360M-Instruct model. The results indicate that cache steering reliably induces the correct structure for common styles such as *Stepwise Reasoning*, *Causal Chain*, and *Analogical Reasoning*. Performance is weaker for less common styles. We provide an analysis of these failure modes in Appendix A.

Figure 4 illustrates outputs from a single ARC-Challenge question under different style-specific steering vectors. These show that the rhetorical differences between generations are not only detectable but often pronounced. For instance, all analogical responses begin with `Just like ...`, while causal chain examples follow a conditional logic pattern. These observations confirm that stylistic signals are indeed encoded in the KV cache and can be carried over to any prompt using cache steering. Taken together, these results show that cache steering can be used not only to induce reasoning in general, but to exert fine-grained control over its *form*.

## 6 CONCLUSIONS

We introduced *cache steering*, a one-shot technique for guiding language models by modifying their key–value cache. Using contrastive examples and GPT-4o-generated reasoning traces, our method induces structured reasoning in small models without fine-tuning, prompt engineering, or continuous interventions. Unlike activation steering, cache steering operates once on fixed past representations, improving stability, efficiency, and compatibility with standard inference pipelines. Experiments on GSM8K, ARC-c, CSQA, and PIQA show reliable induction of reasoning behavior and occasional accuracy gains, while style-transfer experiments demonstrate the ability to control reasoning forms. Although its effectiveness still depends on coefficient settings and steering vector selection, cache steering provides a lightweight, practical mechanism for behavior control and opens new directions for behavior control and low-cost distillation in the KV space.

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

Table 5: **Overview of reasoning styles used in the style transfer experiment**. Each style reflects a distinct structure.

| Style Name | Structure |
| --- | --- |
| Stepwise Reasoning | `Step 1: ... Step 2: ...` |
| Strategy + Execution | `Strategy: ... Solution: ...` |
| Causal Chain | `If ..., then ... Therefore...` |
| Annotated Deduction | `[Premise] → [Inference]` |
| Analogical Reasoning | `This is similar to... Thus, we can infer...` |

## A    ADDITIONAL DETAILS ON STYLE TRANSFER

### A.1    EXPERIMENTAL SETUP

To evaluate whether cache steering can transfer distinct reasoning styles, we select a subset of 20 questions from the ARC-Challenge dataset, each paired with its correct multiple-choice answer. For each question, we construct five distinct reasoning traces that arrive at the same answer but differ in their structure. The description of the five styles can be found in Table 5.

We extract one steering vector per style. During inference, we apply each style-specific steering vector to a set of 20 questions from the test set of the ARC-Challenge dataset to examine how it affects the resulting output structure. For this experiment, we use the SmolLM-360M-Instruct model due to its small size.

### A.2    FAILURE ANALYSIS

Style transfer was less reliable for the *Strategy + Execution* and *Annotated Deduction* formats. Only half of the generations reflected the strategy-execution structure, and just one out of ten matched the annotated deduction style. To understand these failure modes, we performed a qualitative analysis of the outputs. In the case of *Annotated Deduction*, we hypothesize that this format is underrepresented in the model's pretraining distribution. While most of the completions exhibited partial stylistic artifacts, such as starting with a phase or word in square brackets (e.g., `[Farms] in Wyoming were ...`), they lacked the structured logical progression seen in the positive examples used during vector extraction. The steering signal appeared to "nudge" the model in the direction of the desired style, but was not strong enough to elicit full adherence.

A similar pattern emerged with the *Strategy + Execution* format. Although all the responses began with the correct discourse marker (e.g., `Strategy:`), half of the generated samples repeated the same marker in a loop (e.g. `Strategy:, Strategy:, ...`). We attribute this breakdown to possible oversteering: since we did not tune the steering coefficients for each style, it is likely that the default values were too strong in this case, leading to degenerate outputs. These analyses suggest that while cache steering robustly transfers common reasoning styles, rare or structurally complex formats may require further tuning or style-specific adjustments.

## B    QUALITATIVE EXAMPLES

Here we show qualitative examples from different model-dataset pairs. We try to include cases when: 1) both baseline and steering answers are correct; 2) only the steering answer is correct; 3) only the baseline answer is correct. All qualitative examples can be found the the code repository, which contains the outputs for all model-dataset pairs.

Importantly, even though cache steering is effective at inducing reasoning, this alone does not guarantee higher accuracy on the benchmarks. In some cases, it induces reasoning traces that are well-structured and plausible but arrive at incorrect conclusions (e.g., due to question ambiguity or alternative valid interpretations). We also observe that failures or the models to tie complex reasoning back to the choice labels (i.e., generating the correct answer but the wrong label) sometimes cause correct reasoning traces to be scored as incorrect. We encourage readers to examine qualitative outputs in the code repository, where the steering effect is visible in the structure and style of responses.

## B.1 LLAMA-3.2-1B-INSTRUCT ON ARC-C

---

**Llama-3.2-1B-Instruct on ARC-c: both baseline and steering are correct**

**Prompt:** A scientist investigated how well plants native to Maryland would grow in desert conditions. The scientist placed equal amounts of desert soil into 10 identical containers. Each container held two of the same kind of plant. The scientist investigated a total of 10 different kinds of plants. The scientist placed all of the containers into a greenhouse adjusted to produce hot, desert-like conditions. The scientist gave the plants equal, but small amounts of water for the next five weeks. Most of the plants did not survive after the five week investigation. Which statement best explains why 16 of the plants did not survive? Choices:
A: Plants do not grow in desert conditions.
B: Few plants survive severe changes to their environment.
C: Plants native to Maryland are not good for scientific studies.
D: Plants native to Maryland are only able to survive in Maryland.

**Baseline Response:**
The correct answer is B: Few plants survive severe changes to their environment.
The statement that best explains why 16 of the plants did not survive is that they were placed in a desert-like environment with hot, desert-like conditions, and then given small amounts of water for five weeks. This severe change to their environment would likely be stressful for the plants, making it difficult for them to survive. The fact that most of the plants did not survive suggests that the conditions were too harsh for them, rather than that the plants themselves are not good for scientific studies or that they are only able to survive in Maryland. So the correct choice is B.

**Steered Response:**
To understand why 16 of the plants did not survive, we need to consider the factors that affect plant growth in desert conditions.
Firstly, plants native to Maryland are not adapted to desert conditions. They are typically found in temperate climates with moderate temperatures and adequate rainfall. Desert plants, on the other hand, are adapted to hot, dry conditions with limited water availability.
Secondly, the desert soil used in the containers was not suitable for the plants. Desert soil is often high in salt and lacks essential nutrients, which can be detrimental to plant growth.
Thirdly, the amount of water provided to the plants was small, which can be stressful for most plants. Even if the plants were adapted to desert conditions, the lack of water would still be a significant challenge.
Now, let's consider the options:
A: Plants do not grow in desert conditions. This statement is incorrect because plants can grow in desert conditions, but they require specific adaptations and conditions to survive.
B: Few plants survive severe changes to their environment. This statement is partially correct, as the severe changes to the environment (hot, dry conditions and limited water) would indeed be challenging for most plants. However, it does not explain why 16 of the plants did not survive.
C: Plants native to Maryland are not good for scientific studies. This statement is incorrect because plants native to Maryland can be valuable subjects for scientific studies, especially when they are adapted to specific conditions.
D: Plants native to Maryland are only able to survive in Maryland. This statement is incorrect because plants native to Maryland can survive in other environments, including desert conditions, as long as they are adapted to those conditions.
The correct answer is B: Few plants survive severe changes to their environment. This is because the combination of hot, dry conditions and limited water provided to the plants would be challenging for most plants to survive. So the correct choice is B.

---

**Llama-3.2-1B-Instruct on ARC-c: incorrect baseline, correct steering**

**Prompt:** A college campus has so few parking spaces that cars are often lined up waiting to park during class hours. Which of the following describes how the college could best solve the need for more parking spaces using the universal systems model? Choices:
A: The college could build an additional parking lot, gather parking data, and then receive input from community meetings.
B: The college could limit access to parking lots during busy hours, hire security officers to help restrict parking, and then gather parking data.
C: The college could gather parking data, receive input from community meetings, build an additional parking lot, and then gather more parking data.
D: The college could hire security officers to help restrict parking, gather parking data, receive input from community meetings, and then gather more data.

**Baseline Response:**
The correct answer is B: The college could limit access to parking lots during busy hours, hire security officers to help restrict parking, and then gather parking data.
This option describes how the college could best solve the need for more parking spaces using the universal systems model. The universal systems model is a framework for understanding and managing complex systems, and in this case, it involves:
Identifying the problem (limited parking spaces)
Gathering data on the current situation (parking data)
Receiving input from stakeholders (community meetings)
Taking action to address the problem (limiting access to parking lots during busy hours and hiring security officers)
This approach is a good example of how the universal systems model can be applied to solve a complex problem like limited parking spaces. So the correct choice is B.

**Steered Response:**
To solve the need for more parking spaces on a college campus, the college should consider the universal systems model, which involves several key components.
First, the college should gather parking data. This involves collecting information about the number of cars parked, the types of vehicles, and the time of day when parking is most popular. This data will help the college understand the current parking situation and identify areas where additional parking spaces are needed.
Next, the college should receive input from community meetings. This involves engaging with students, faculty, and staff to gather their opinions and concerns about parking. By listening to their feedback, the college can better understand the needs and priorities of the community.
Building an additional parking lot is a crucial step in addressing the parking shortage. However, it should be done in conjunction with the gathering of parking data and the community meetings. This ensures that the new parking lot is designed and constructed to meet the needs of the community, and that it is accessible and convenient for everyone.
Finally, the college should gather more parking data to evaluate the effectiveness of the new parking lot and make adjustments as needed. This will help the college to refine its parking management strategies and ensure that the parking lot is meeting its intended purpose.
In summary, the correct answer is the one that involves gathering parking data, receiving input from community meetings, building an additional parking lot, and then gathering more data to evaluate its effectiveness. Choices A, B, and D do not accurately describe the universal systems model. Choice A involves gathering data and then implementing a solution, but it does not involve community meetings. Choice B involves restricting access to parking during busy hours, but it does not involve gathering data or receiving input from the community. Choice D involves gathering data, but it does not involve building an additional parking lot or receiving input from the community. So the correct choice is C.

---

### Llama-3.2-1B-Instruct on ARC-c: correct baseline, incorrect steering

**Prompt:** In 1903 Mary Anderson invented the first windshield wiper. How did this invention most likely help people?
Choices:
A: It helped people use less gas.
B: It kept people from driving too fast.
C: It made cars easier for people to buy.
D: It made cars safer to drive in bad weather.

**Baseline Response:**
The correct answer is D: It made cars safer to drive in bad weather.
Mary Anderson's windshield wiper invention was designed to help people see while driving, especially in rainy or snowy conditions. By keeping the windshield clear of rain and snow, drivers could see the road ahead more clearly, reducing the risk of accidents. This was particularly important in the early days of automobile travel, when roads were often poorly maintained and visibility was limited. Anderson's invention helped to improve road safety and reduce the number of accidents on the roads. So the correct choice is D.

**Steered Response:**
To understand how Mary Anderson's windshield wiper invention helped people, we need to consider the context and purpose of the invention.
Mary Anderson's windshield wiper was designed to improve visibility while driving, especially in rainy or foggy conditions. The primary goal of the invention was to reduce the amount of time spent stopping and starting the vehicle, which was a significant challenge in early automobiles.
In the early 1900s, driving was a hazardous activity, especially in poor weather conditions. Drivers had to constantly stop and start the engine, which was time-consuming and often resulted in accidents. The windshield

wiper helped to reduce this time by allowing drivers to clear the windshield of rain and debris more quickly. Therefore, the correct answer is that Mary Anderson's windshield wiper invention most likely helped people by reducing the time spent stopping and starting the vehicle, which was a significant challenge in early automobiles. So the correct choice is B.

## C   IMPLEMENTATION DETAILS

### C.1   CONTRASTIVE DATA CONSTRUCTION

All prompts are formatted using the chat template from the model's tokenizer configuration. This ensures consistency with how the models are typically used during chat-style inference. The example of a single contrastive pair with 1 in-context example:

Figure 5: **The example of a single contrastive pair with 1 in-context example**. The positive example (left) includes both the reasoning trace and the final answer. The negative example (right) includes only the final answer.

To ensure that in-context examples are semantically similar to the target question, we embed all training questions using the `all-MiniLM-L6-v2` sentence embedding model (Reimers & Gurevych, 2019) and retrieve the top-$n$ most similar examples based on cosine similarity.

### C.2   ANSWER EXTRACTION

We explored using an LLM to extract answer labels from model outputs, which is a common practice in recent activation steering studies (Rimsky et al., 2024; Wang et al., 2023; Wehner et al., 2025). However, due to computational constraints, we were unable to use a sufficiently large model to ensure high-quality extraction. In particular, smaller judges often rely on their own knowledge to infer the correct answer, rather than faithfully extracting it from the generated output. This compromises the reliability of evaluation in cases where steering affects reasoning without necessarily correcting the final answer. To minimize the amount of false positive answers, we opted for a rule-based pipeline that is transparent, fast to run at scale, and robust enough for comparative analysis in our setting.

**GSM8K.**   For GSM8K, we extract the final numeric answer using a digit-based pattern match. Specifically, we select the last number mentioned in the model's output. This approach has been used in prior work (Wang et al., 2023; Wang & Zhou, 2024). While this method introduces both false positives (e.g., trailing numbers in explanations) and false negatives (e.g., answers embedded in text), these effects tend to cancel out over large-scale evaluation.

**Multiple-choice tasks.**   For multiple-choice datasets, we require more structured extraction due to the open-ended nature of the model outputs. Therefore, we adopt the approach from Wang & Zhou (2024) and augment it further with a more rigorous extraction process. We develop a multi-stage extraction pipeline designed to recover answer labels with high precision and robustness. For all

experiments, we append the string `"So the correct choice is"` to the end of the prompt and allow the model to generate 10 additional tokens. The output is then processed in the following stages:

1. **Regex-based label extraction:** We apply a sequence of 12 regular expressions to identify explicit label mentions (e.g., "A", "option C", "(B)") both immediately following the answer prefix and in the whole answer string as a fallback. Each pattern is designed to handle common formats seen across models and datasets. The full list of regex patterns can be found in the code.

2. **String match fallback:** If no regex match is found, we scan the span after `"So the correct choice is"` for an exact string match with any of the raw answer choices. To avoid false positive matches, we only accept matches when exactly one choice matches unambiguously. We also filter out cases containing negation cues (e.g., "not", "incorrect", "wrong") to exclude completions like "So the correct answer is not B".

3. **Invalid and multi-label detection:** We detect multi-answer completions (e.g., "both A and C") or noncommittal outputs ("none of the above") and label them as `[incorrect]`. Any remaining answers that do not match a valid choice are marked as `[invalid]`.

4. **Constrained decoding for fallback resolution:** For all completions marked as `[invalid]`, we discard the answer span after `"So the correct choice is"` and repeat decoding with constraints: we re-append `"So the correct choice is"` to the prompt and sample a single token, masking the logits to allow only valid label tokens. This final step guarantees that a valid label is recovered for every input.

Despite this multi-stage approach, errors in label extraction are still common. In particular, semantically correct answers may be incorrectly marked due to minor phrasing differences or ambiguous generation formats. Additionally, the constrained decoding stage forces the model to generate a valid label even if the generated text is not semantically meaningful, which was especially common in activation steering experiments. Therefore, to mitigate this, we discard the results if the accuracy before the constrained decoding stage is significantly lower than after.

### C.3 ACTIVATION STEERING EVALUTAION

To make a fair comparison of activation steering to cache steering, we chose a similar setting: greedy decoding, 200 contrastive samples, each with 5 few-shot examples. Similarly to cache steering implementation, the vectors were extracted from the last token position, aggregated with a Difference-in-Means method. The vectors were applied continuously to each new token during decoding. All these choices adhere to the current best practices in activation steering research (Wehner et al., 2025). To extract and apply the steering vector, we used the steering-vectors Python library (Chanin, 2025) that implements the most popular activation steering method CAA (Rimsky et al., 2024).

First we performed a grid search on a subset of data over $c \in [0.5, 1, 3]$ and middle layers of each model: for SmolLM2-360M-Instruct $l \in [13, 14, 15, 16, 17, 18, 19]$, for Llama-3.2-1B-Instruct $l \in [6,7,8,9,10]$, for Llama-3.2-3B-Instruct $l \in [13,14,15]$, for Llama-3.1-8B-Instruct $l \in [15,16,17]$, for Phi-4-mini-instruct $l \in [15, 16, 17]$, for Qwen2-0.5B-Instruct $l \in [11, 12, 13]$. The selected hyperparameters were inspired by the numbers reported in Turner et al. (2024). Then, activation steering with the best parameters was evaluated on the full test set to obtain the final results.

### C.4 SAMPLING PARAMETERS

For sampling experiments, we used the parameters specified in the generation config of the model. If the generation config was not available, we used temperature: 0.6, top_p: 0.9, top_k: 50.

### C.5 ALIGNING CACHE POSITION

Through extensive experimentation, we discovered that for most datasets, cache steering is most effective if the vectors are applied to the same token from which the vectors were extracted. In case of an instruction-tuned model, such a token can be the last token of the generation prompt (e.g., the newline character after "assistant"). Therefore, when we extract the steering vectors from such a token, the steering effect is most pronounced if we apply the vector to the same token. Even though the actual positions of the extraction and application tokens in the corresponding sequences are different, in

both sequences the tokens play a similar role. We think of these tokens as information aggregation tokens. In cases when the desired application token is last in the sequence, we append a special token to the prompt in order to be able to apply the intervention to the correct position. In cases when cache steering is applied to any other position, this procedure is not needed.

### C.6 GENERATION OF REASONING DATA

To generate reasoning data, we used a GPT-4o model via OpenAI's Chat Completions API. We used the following instruction prompt to elicit detailed CoT-style responses:

```
You are given a question and a corresponding answer
to that question.  Your task is to think step by
step and provide the reasoning steps to get the
answer.  Separate each reasoning step with <reasoning>
</reasoning> tags.  The question:  '{question}'.  The
correct answer:  {answer}.
```

The obtained reasoning steps were further parsed with regular expressions.

### C.7 COMPUTATIONAL OVERHEAD EXPERIMENT DETAILS

To compare the computational efficiency of cache steering, activation steering, and the model without any intervention, we measure the per-token generation time under both methods using a subset of 100 examples from the ARC-Challenge dataset.

We conduct experiments using two different batch sizes: 1 (single-example inference) and 16 (batched inference), to reflect both interactive and throughput-oriented use cases. All runs are executed on the same hardware using greedy decoding.

Timing is measured from the beginning of generation (post-prompt forward pass) to the completion of the final token. All results are averaged over three runs.

### C.8 MODELS USED

We evaluate our method using multiple open-source language models from different model families. Below, we list their Hugging Face model hub URLs.

- **SmolLM2-360M-Instruct**
  - URL: https://huggingface.co/HuggingFaceTB/SmolLM2-360M-Instruct
  - License: Apache 2.0
- **LLaMA-3.2-1B-Instruct**
  - URL: https://huggingface.co/meta-llama/Llama-3.2-1B-Instruct
  - License: Llama 3.2 Community License
- **LLaMA-3.2-3B-Instruct**
  - URL: https://huggingface.co/meta-llama/Llama-3.2-1B-Instruct
  - License: Llama 3.2 Community License
- **Qwen2-0.5B-Instruct**
  - URL: https://huggingface.co/Qwen/Qwen2-0.5B-Instruct
  - License: Apache 2.0
- **Phi-4-mini-instruct**
  - URL: https://huggingface.co/microsoft/Phi-4-mini-instruct
  - License: MIT

- **Llama-3.1-8B-Instruct**
  - URL: https://huggingface.co/meta-llama/Llama-3.1-8B-Instruct
  - License: Llama 3.1 Community License

## D  REPRODUCIBILITY

### D.1  EXPERIMENTS REPRODUCIBILITY

To ensure consistency and facilitate detailed analysis, we implemented several reproducibility safeguards throughout our experimental pipeline.

**Sample tracking via UUIDs.**  At test time, we assign each input a unique identifier (UUID) derived deterministically from a hash of the input text. This allows us to track and compare individual examples across experiments with different settings (e.g., sampling, steering variants, decoding strategies), and ensures the integrity of input data over time. The UUID makes it easy to locate the same question across logs, qualitative outputs, and evaluation reports, and is sensitive to minor changes in the question itself.

**Deterministic runs.**  For all runs involving stochastic generation (e.g., sampling-based decoding), we set the random seed at the beginning of each run to guarantee reproducibility.

**Llama chat template**  In all experiments, we used the chat template predefined by the model to tokenize the input text. However, we noticed that specifically in the Llama models, the current date is added to the system prompt, making it impossible to fully reproduce the results. Therefore, we modify the chat template of the Llama models to exclude the current date from the system prompt.

### D.2  HARDWARE SPECIFICATIONS

All experiments were run on the internal cluster (not in the cloud). These are the specifications of the hardware:

- 1 NVIDIA H100 GPU, 94GiB of memory
- 16 AMD 4th GEN EPYC CPUs

The time to run each experiment varied per model, dataset, whether it was a baseline experiment, cache steering or activation steering experiment, the amount of training data used, etc. On average, a single run for almost all model-dataset pairs took less than 1 hour to run, with the exception of Llama-3.2-3B model on PIQA dataset, which took under 2 hours, and activation steering experiments, which took under 6 hours per experiment. The full research project required more compute than the experiments reported in the paper since a significant part of the project was experimentation and empirical analysis.

### D.3  SOFTWARE ENVIRONMENT

- Python 3.11.11
- transformers: 4.49.0
- torch: 2.5.1

## E  SENSITIVITY OF ACTIVATION STEERING TO HYPERPARAMETERS

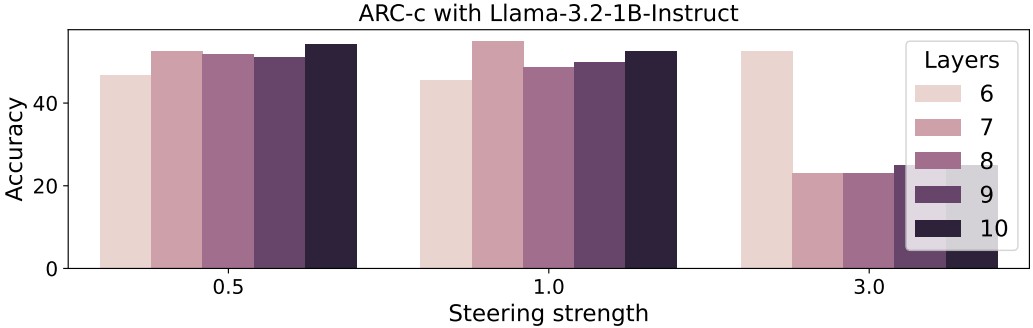

Figure 6: Sensitivity of activation steering to hyperparameters on ARC-c dataset using Llama-3.2-1B. The results are obtained from the activation steering grid search described in C.3.

# F  LENGTH OF GENERATED OUTPUTS

Table 6: **Cache steering consistently increases the length of generated outputs across tasks**. We report the average number of generated tokens under three conditions: baseline decoding, CoT prompting, and cache steering. Results are shown for multiple model sizes across four reasoning benchmarks. The results indicate that cache steering leads to longer answers on average across all tasks and models, except for GSM8K. We hypothesize that the reason for that is that this dataset is a classic benchmark for evaluation of reasoning methods, and all models are already trained on this dataset and generate CoT responses even without explicit instructions.

| Task | Model | Baseline | CoT | Cache Steering |
|------|-------|----------|-----|----------------|
| ARC | SmolLM2-360M-Instruct | 22 | 188 | **315** |
| | Qwen2-0.5B-Instruct | 75 | 147 | **242** |
| | Llama-3.1-8B-Instruct | 178 | 196 | **315** |
| | Llama-3.2-1B-Instruct | 152 | 171 | **283** |
| | Llama-3.2-3B-Instruct | 190 | 208 | **301** |
| | Phi-4-mini-instruct | 110 | 228 | **369** |
| CSQA | SmolLM2-360M-Instruct | 19 | 159 | **295** |
| | Qwen2-0.5B-Instruct | 53 | 104 | **181** |
| | Llama-3.1-8B-Instruct | 104 | 151 | **340** |
| | Llama-3.2-1B-Instruct | 46 | 134 | **283** |
| | Llama-3.2-3B-Instruct | 105 | 146 | **251** |
| | Phi-4-mini-instruct | 62 | 181 | **443** |
| GSM8K | SmolLM2-360M-Instruct | 214 | **222** | 222 |
| | Qwen2-0.5B-Instruct | 187 | **216** | 188 |
| | Llama-3.1-8B-Instruct | 185 | 179 | **207** |
| | Llama-3.2-1B-Instruct | 147 | 146 | **150** |
| | Llama-3.2-3B-Instruct | 171 | 167 | **179** |
| | Phi-4-mini-instruct | 142 | **215** | 137 |
| PIQA | SmolLM2-360M-Instruct | 39 | 210 | **345** |
| | Qwen2-0.5B-Instruct | 85 | 204 | **289** |
| | Llama-3.1-8B-Instruct | 157 | 173 | **327** |
| | Llama-3.2-1B-Instruct | 142 | 196 | **449** |
| | Llama-3.2-3B-Instruct | 175 | 203 | **407** |
| | Phi-4-mini-instruct | 117 | 220 | **366** |

# G    LIST OF HYPERPARAMETERS

Table 7: Hyperparameters used for the experiments for each task-dataset pair.

| Task | Model | Contrastive Samples | In-context Examples | $c^k$ | $c^v$ |
|---|---|---|---|---|---|
| GSM8K | HuggingFaceTB/SmolLM2-360M-Instruct | 200 | 5 | 0 | 1 |
| | meta-llama/Llama-3.2-1B-Instruct | 100 | 5 | 0 | 1 |
| | meta-llama/Llama-3.2-3B-Instruct | 100 | 5 | 0 | 1 |
| | Qwen/Qwen2-0.5B-Instruct | 100 | 5 | 0 | 3 |
| | meta-llama/Llama-3.1-8B-Instruct | 100 | 5 | 0 | 2 |
| | microsoft/Phi-4-mini-instruct | 100 | 5 | 0 | 1 |
| CSQA | meta-llama/Llama-3.2-1B-Instruct | 100 | 5 | 0 | 10 |
| | meta-llama/Llama-3.2-3B-Instruct | 300 | 10 | 0 | 4 |
| | HuggingFaceTB/SmolLM2-360M-Instruct | 400 | 12 | 0 | 6 |
| | Qwen/Qwen2-0.5B-Instruct | 200 | 10 | 0.2 | 4 |
| | meta-llama/Llama-3.1-8B-Instruct | 100 | 5 | 0 | 10 |
| | microsoft/Phi-4-mini-instruct | 100 | 5 | 0 | 10 |
| ARC-c | meta-llama/Llama-3.2-3B-Instruct | 400 | 10 | 0 | 6 |
| | meta-llama/Llama-3.2-1B-Instruct | 200 | 10 | 0 | 6 |
| | HuggingFaceTB/SmolLM2-360M-Instruct | 300 | 10 | 0 | 6 |
| | Qwen/Qwen2-0.5B-Instruct | 400 | 10 | 0 | 10 |
| | meta-llama/Llama-3.1-8B-Instruct | 200 | 10 | 0 | 6 |
| | microsoft/Phi-4-mini-instruct | 200 | 10 | 0 | 6 |
| PIQA | meta-llama/Llama-3.2-1B-Instruct | 200 | 10 | 0 | 6 |
| | meta-llama/Llama-3.2-3B-Instruct | 200 | 10 | 0 | 10 |
| | HuggingFaceTB/SmolLM2-360M-Instruct | 200 | 10 | 0 | 6 |
| | Qwen/Qwen2-0.5B-Instruct | 200 | 10 | 0 | 8 |
| | meta-llama/Llama-3.1-8B-Instruct | 200 | 10 | 0 | 6 |
| | microsoft/Phi-4-mini-instruct | 200 | 10 | 0 | 6 |

# H    LIMITATIONS

In this work, we focus primarily on inducing reasoning behavior in small LLMs. While results on one larger model already show a potential for even better improvements, further study is needed to assess how well cache steering generalizes across a wider range of large models, domains, and tasks beyond reasoning. Importantly, cache steering, like other behavior-guidance methods including prompting and fine-tuning, has broad applications. Potential misuse, such as steering toward deceptive, harmful, or biased outputs, remains a concern. We therefore advocate responsible use and recommend that safeguards are considered.

# I  VERTICAL AMPLIFICATION

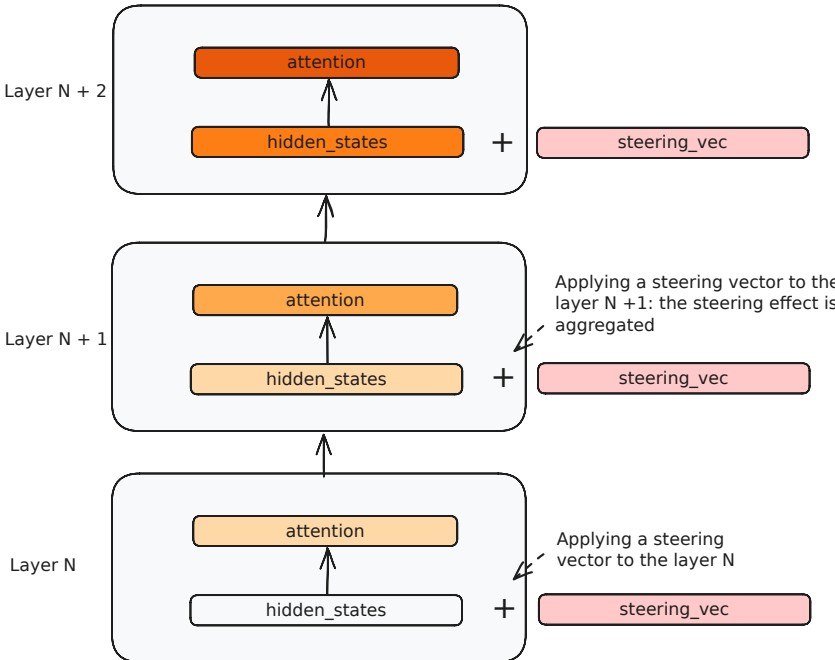

Figure 7: **Vertical amplification in activation steering.** A steering vector added to the residual stream at layer $N$ influences the representations of each subsequent layer, causing the steering effect to accumulate vertically. Applying anoter steering vector at layer $N+1$ reinforces amplifies the the steering effect, leading to oversteering. This vertical propagation contrasts with cache steering, where modifications are applied to static key-value vectors that are not recomputed across layers.

# J ADDITIONAL EXPERIMENTS

## J.1 ABLATION ON INDIVIDUAL LAYERS

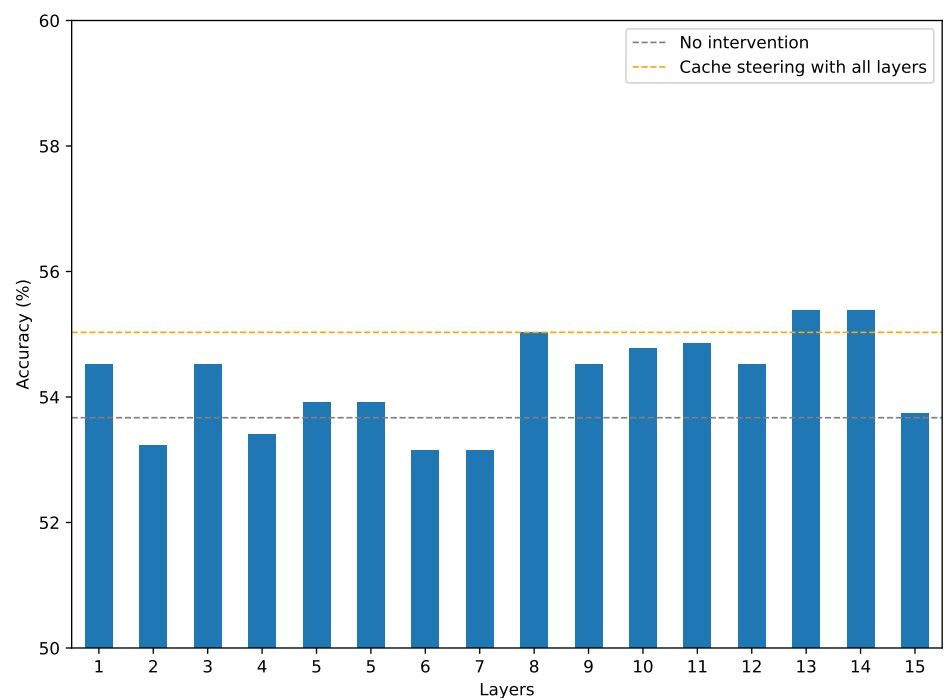

Figure 8: **Ablation on the individual layers on ARC-c (Llama-3.2-1B-Instruct).** The results show only modest variation ($\approx 2\%$ points) across layers, supporting our claim of reduced sensitivity compared to activation steering.

To further examine the sensitivity of cache steering to the choice of steering layers, we ran an ablation in which the steering vector was applied to each individual layer of Llama-3.2-1B-Instruct on ARC-c. Figure 8 presents the accuracy obtained when steering only a single layer at a time. The results vary by only $\approx 2$ percentage points across all layers. This behavior contrasts with activation steering, where effectiveness depends heavily on choosing the correct layer and where a poor choice can lead to degraded performance. Overall, this ablation supports our claim that cache steering is substantially less sensitive to the steering location, reducing the hyperparameter search space to steering strength coefficients.

## J.2 PERSISTENCE OF EFFECT OVER LONG GENERATIONS

To evaluate whether the steering effect persists over long generations, we conduct a targeted analysis on Llama-3.2-1B-Instruct using the ARC-c dataset. For each input, we first apply cache steering for the initial $n$ generated tokens (with $n \in \{0, 16, 32, 64, 128, 256, 512, 1024\}$) and then reset the KV cache, removing all steering modifications before continuing decoding. We measure the final response length and structure over 100 samples per setting. The results, shown in Figure 9, demonstrate a clear monotonic trend: the longer steered KV cache is used, the longer and the final responses remain. When conditioned only on the previously generated tokens (without modified KV cache), the model tends to produce shorter sequences.

## J.3 T-SNE VISUALIZATION OF POSITIVE AND NEGATIVE VALUE REPRESENTATIONS

To better understand why contrastive cache shifts are effective we follow the methods used by (Azizi et al., 2025; Chen et al., 2025), we visualize the value vectors extracted from positive (reasoning) and negative (non-reasoning) examples using 2-D t-SNE projections (Maaten & Hinton, 2008) across all layers of Llama-3.2-1B-Instruct on ARC-c (Figure 10). We observe consistent separation between

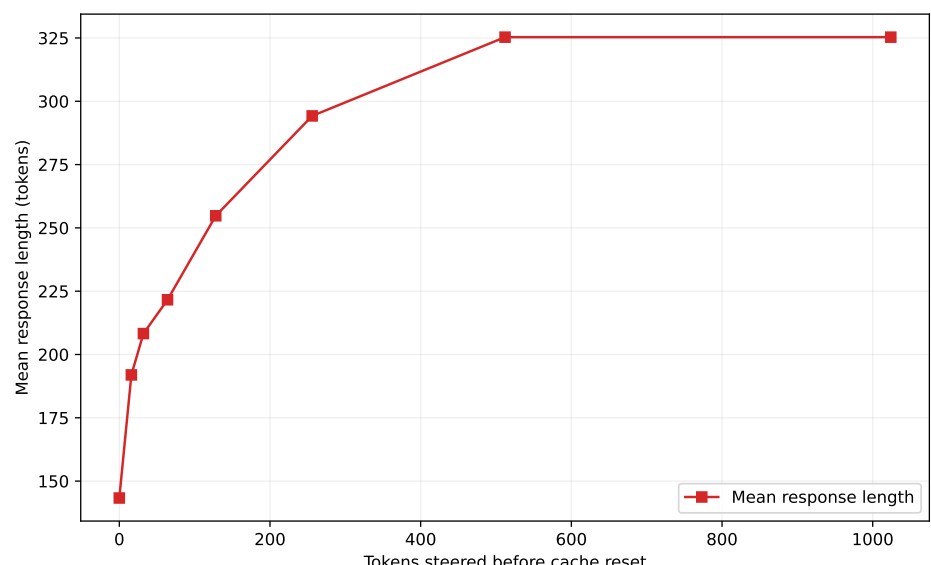

Figure 9: **Persistence of cache steering effect over long generations.** Mean response length on ARC-c for Llama-3.2-1B-Instruct when steered KV cache is used to generate only the first $n$ tokens (x-axis), after which the KV cache is reset and decoding proceeds normally ($n \in \{0, 16, 32, 64, 128, 256, 512, 1024\}$).

positive and negative representations in almost every layer, indicating that the model internally encodes reasoning-related signals in a linearly separable manner. This directly motivates our method: constructing a steering vector as the difference of mean positive and negative cache representations captures this separation and provides a direction that reliably induces reasoning behavior.

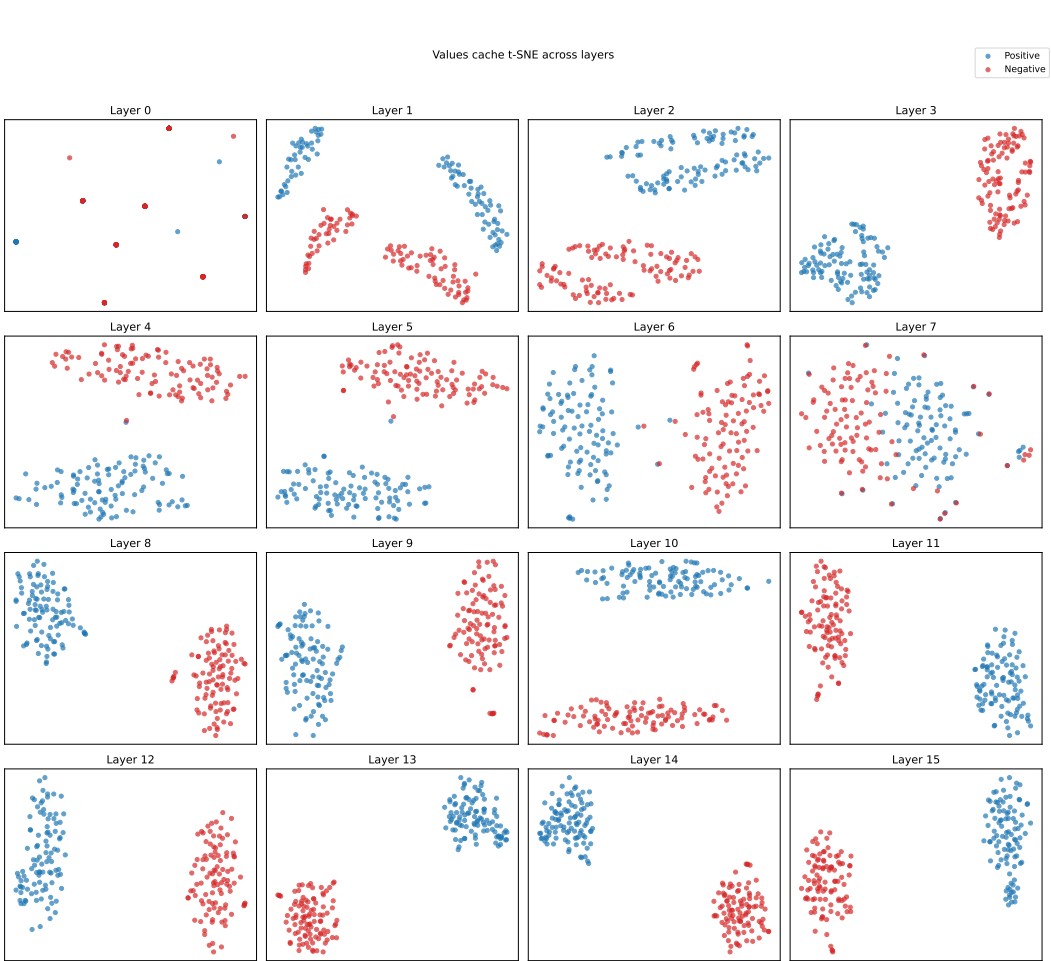

Figure 10: **t-SNE visualization of the value representations of positive and negative examples in each layer on ARC-c (Llama-3.2-1B-Instruct).** Two-dimensional t-SNE projections of 100 positive (reasoning) and 100 negative (non-reasoning) value-cache vectors from Llama-3.2-1B-Instruct on ARC-c. Almost all layers show clear separation between the two groups, indicating that reasoning-related signals are encoded distinctly in the value cache and motivating the construction of steering vectors via contrastive averaging in the KV space.

