# OpenReview forum: "KV Cache Steering for Controlling Frozen LLMs"
_ICLR.cc/2026/Conference — Submitted to ICLR 2026_

### Official Review · Reviewer_qbMb · 2025-10-31

**Soundness:** 2
**Presentation:** 4
**Contribution:** 2
**Rating:** 4
**Confidence:** 5

**Summary:**

This paper proposes cache steering, which, unlike previous Activation Steering, applies (patches) steering to the KV cache instead of each hidden activations (last position). According to the authors, this reduces computational overhead and sensitivity to hyperparameters. The core idea is to apply a one-shot modification to the prefilled key–value cache using steering vectors extracted from reasoning traces generated by a teacher model or human annotations. Experimental results show that cache steering improves both the reasoning structure and task accuracy across multiple small and medium-sized models, outperforming activation steering while incurring virtually no additional latency.

**Strengths:**

- The proposed cache steering is computationally more efficient than activation steering by removing the need for continuous activation editing.
- The paper provides comprehensive evaluations across multiple models and datasets, with well-designed ablations on contrastive pair size, steering strength, and reasoning style.
- The writing is clear, and the figures and tables are well-organised, improving readability and accessibility.

**Weaknesses:**

If I have misunderstood any of the following points or if additional evidence can be provided, I would happily reconsider my evaluation.

- The central idea of this work is to apply steering to the KV cache rather than to the hidden activations. However, as noted by the authors themselves in the related work section (Liu et al., 2025b), prior studies have already explored KV-cache-based interventions for enhancing reasoning ability. It would therefore be helpful if the authors could clarify what novel research insight or conceptual contribution distinguishes this paper from those earlier approaches. While the proposed idea does not necessarily need to be complex, it should clearly articulate its unique conceptual advance beyond existing work.
- The paper discusses the over-steering problem, but this issue has already been addressed in prior work through Dynamic Steering (e.g., Scalena et al., 2024 [1]; Do et al., 2025 [2]). A comparison with these approaches is needed in my opinion.
- The claim that KV-cache-based steering is less sensitive to hyperparameter choices requires stronger evidence. The authors only vary the number of contrastive pairs, few-shot examples, and steering strengths. In the experiments, cache steering is applied to all layers and specifically to a neutral offset token. It remains unclear whether the method would remain robust if applied to different layers or different token positions (see questions below).

[1] Multi-property Steering of Large Language Models with Dynamic Activation Composition, Scalena et al., 2024, Proceedings of the 7th BlackboxNLP Workshop: Analyzing and Interpreting Neural Networks for NLP

[2] Dynamic Steering With Episodic Memory For Large Language Models, Do et al., 2025, Findings of the Association for Computational Linguistics: ACL 2025

**Questions:**

- Is KV-cache-based steering truly less sensitive to steering location as a hyperparameter? If applying it to only specific layers or to arbitrary token positions instead of the neutral offset token produces better results, then the method would still require grid search, similar to activation steering.
- When steering is applied at each layer, doesn’t this still risk over-steering, since it modifies the representation at multiple points, analogous to injecting steering vectors into each residual stream? (although I agree that the over-steering would be less than the original activation steering)
- In Table 1, cache steering underperforms activation steering on GSM8K for three models. Can the authors explain this behaviour? Also, have they tested a combination of activation steering + CoT prompting for a fair comparison?
- As the generation length increases, the influence of the cache intervention may decay. Did the authors analyse this effect? If it does weaken over long generations, have they considered combining their method with continuous patching (as in standard activation steering)?

---

> ### Author Response · Authors · 2025-11-21
>
> Dear Reviewer,
>
> Thank you for the thoughtful feedback and for recognizing the computational efficiency of our method, the breadth of our evaluations and ablations, and the clarity of our presentation. Below, we address the reviewer’s questions and provide additional clarification.
>
> ---
>
> > “The central idea of this work is to apply steering to the KV cache rather than to the hidden activations. However, as noted by the authors themselves in the related work section (Liu et al., 2025b), prior studies have already explored KV-cache-based interventions for enhancing reasoning ability. It would therefore be helpful if the authors could clarify what novel research insight or conceptual contribution distinguishes this paper from those earlier approaches. While the proposed idea does not necessarily need to be complex, it should clearly articulate its unique conceptual advance beyond existing work.”
>
> We thank the reviewer for raising this point. While Liu et al. (2025b) also operate on KV caches, their *Coprocessor* approach requires **training a separate LLM** to *generate* and *concatenate* additional KV vectors into the cache, effectively a learned augmentation module. In contrast, **our method requires no training, no auxiliary models, and no architectural modifications**: we simply add a steering direction to existing KV entries as a single post-prefill edit, after which decoding proceeds normally.
>
> We already discuss this distinction in lines 148–153, but we will try to make it more explicit in the revision. The conceptual contribution of our work is to show that a simple, training-free KV-cache shift is sufficient to induce and control reasoning behavior.
>
> > “The paper discusses the over-steering problem, but this issue has already been addressed in prior work through Dynamic Steering (e.g., Scalena et al., 2024 [1]; Do et al., 2025 [2]). A comparison with these approaches is needed in my opinion.”
>
> We thank the reviewer for raising these works and will cite them in the revision. Dynamic steering methods such as Dynamic Activation Composition (Scalena et al., 2024) and Episodic-Memory Steering (Do et al., 2025) address oversteering by continuously adapting activation edits during decoding. These approaches are **substantially more complex** – e.g., DAC requires two forward passes per generation step to compute KL-guided adjustments, and episodic-memory steering maintains memory banks and per-chunk interpolations between global and local vectors.
>
> Our method is complementary and focuses on **practical simplicity**. By intervening on the KV cache rather than the residual stream, we apply a single post-prefill modification that does not propagate vertically through layers, leading to a stable effect (as supported by Figure 2 and Appendix E for comparison).
>
> Crucially, this simplicity makes cache steering directly usable in real inference stacks (e.g., llama.cpp, Ollama, vLLM) without modifying the decoding loop or model internals. In this sense, our method makes it actually feasible to apply steering in production settings, bridging the gap between research and real-world usage.
>
> > “The claim that KV-cache-based steering is less sensitive to hyperparameter choices requires stronger evidence. The authors only vary the number of contrastive pairs, few-shot examples, and steering strengths. In the experiments, cache steering is applied to all layers and specifically to a neutral offset token. It remains unclear whether the method would remain robust if applied to different layers or different token positions (see questions below).”
>
> We applied cache steering to all layers and to a single empirically selected aggregation token (the final generation prompt token). Importantly, *the offset token is used only to offset the cache positions* to be able to apply the intervention to the correct token (due to how KV caching is implemented) – we do not apply steering to the offset token. We chose this configuration because it worked robustly across all tested models without requiring layer search – one of the practical advantages of cache steering. Prior studies have noted that such aggregation tokens naturally concentrate semantic information, which motivated our choice.
>
> We also agree that additional evidence is helpful. In the revision, we include an ablation on ARC-c with Llama-3.2-1B varying where the intervention is applied and compare it to the baseline and intervention on all layers. The results show only modest variation (≤~2% points) across choices, supporting our claim of reduced sensitivity compared to activation steering. Importantly, this provides additional empirical evidence that with cache steering we can just apply the intervention to all layers, which makes the method more practical.
>
> We add the results of this additional ablation study to the manuscript in Appendix J.1.

---

> ### Author Response · Authors · 2025-11-21
>
> > “Is KV-cache-based steering truly less sensitive to steering location as a hyperparameter? If applying it to only specific layers or to arbitrary token positions instead of the neutral offset token produces better results, then the method would still require grid search, similar to activation steering.”
>
> In principle, one could search over layers or token positions. However, a key advantage of cache steering is that this is not necessary in practice. Our experiments show that applying the steering vector to all layers and to a single aggregation token **works robustly across models and datasets, without layer selection or position tuning**.
>
> While one still needs to choose a steering strength, this is two scalar hyperparameters, and our ablations show that performance is not highly sensitive to their value as opposed to activation steering, where local variations in steering strength lead to drastic performance drops.
>
> > “When steering is applied at each layer, doesn’t this still risk over-steering, since it modifies the representation at multiple points, analogous to injecting steering vectors into each residual stream? (although I agree that the over-steering would be less than the original activation steering)”
>
> We would like to clarify that applying cache steering at all layers is **not analogous** to injecting steering vectors into each residual stream. In activation steering, edits to the residual stream at earlier layers affect every subsequent layer, which leads to compounding and oversteering if the intervention is applied to more than 1 layer (we added a diagram to illustrate this in Appendix I in the revised manuscript).
>
> In contrast, **KV-cache entries for past tokens are *fixed* once computed**. Modifying $K_l$, $V_l$ at layer $l$ does not alter the representations at layer $l+1$, since those KV values are never recomputed. The intervention therefore *propagates only horizontally*, via attention over the cache, rather than vertically through the network.
>
> We believe that this property is a central reason why cache steering remains stable even when applied to all layers, as we describe in lines 221–226.
>
> > “In Table 1, cache steering underperforms activation steering on GSM8K for three models. Can the authors explain this behaviour? Also, have they tested a combination of activation steering + CoT prompting for a fair comparison?”
>
> Since many instruction-tuned models already produce CoT-like outputs on GSM8K without prompting, likely due to widespread use of this dataset for training and evaluation, so we believe that further “reasoning induction” can have *diminishing returns*. We can clearly see this difference with other datasets in Table 6 (Appendix F), which shows the length of generated outputs for each task separately. There we can see that GSM8K is the only dataset where cache steering doesn't consistently result in longer responses, as well as that the baseline responses (even without CoT prompting) are already longer than for other datasets.
>
> Regarding activation steering + CoT, indeed, it would be interesting to see the results. However, since activation steering incurs significant computational overhead due to continuous interventions during decoding, the activation steering experiments take significantly more time and compute. Therefore, due to limited academic compute, we did not perform these experiments as they would not change the overall conclusion – that cache steering works. Instead, we focused on additional experiments that further strengthen our claims about the effectiveness of cache steering.
>
> > “As the generation length increases, the influence of the cache intervention may decay. Did the authors analyse this effect? If it does weaken over long generations, have they considered combining their method with continuous patching?”
>
> We thank the reviewer for the insightful question. While we did not explicitly analyze decay in the initial submission, several results already suggest that the steering effect remains stable: Table 2 shows that cache steering reliably produces longer reasoning sequences than CoT prompting, which is indirect evidence that the induced behavior persists throughout generation.
>
> In the revision, we add a more targeted analysis. Specifically, we generate the first $n$ tokens under cache steering, then reset the KV cache (removing the steering effect) and continue decoding to the end. Comparing the before/after distributions of reasoning length and structure across varying $n$ allows us to measure whether the effect weakens over time. We add the results of this experiment to Appendix J.3 in the revised manuscript. The **results indicate that the steering influence remains stable over long generations**. Given this stability, we have not found a need to combine cache steering with continuous activation patching.
>
> ---
>
> We hope we have clarified all the concerns; if there are no remaining questions, we would kindly ask the reviewer to update the score.

---

### Official Review · Reviewer_Pttn · 2025-11-01

**Soundness:** 3
**Presentation:** 3
**Contribution:** 2
**Rating:** 4
**Confidence:** 5

**Summary:**

The paper proposes a KV cache steering method to induce reasoning capability to LLMs by modifying its internal hidden states. In specific, the KV steering is applied as a one time modification of the prefill KV and the steering is computed by following the reasoning traces of a capable teacher model like GPT-4o. The authors claim that compared to activation steering, which applies interventions at every decoding step, cache steering avoids cascading effects, is robust to hyperparameter choices, introduces virtually no runtime cost, and seamlessly integrates with standard inference pipelines.

**Strengths:**

1. Beyond simple reasoning induction, KV cache steering enables controllable transfer of reasoning styles, without requiring any computation heavy re-training of the model.

2. The KV cache steering is an one time effort compared to that with activation steering that is a repetitive effort (applies during each decode state) and requires fewer hyperparameter tuning overhead.

**Weaknesses:**

1. While this work of leveraging steering vector to induce reasoning thoughts in a model is different, there exists other steering method that are applied to reduce the reasoning thoughts (example: [1, 2]).  Thus it is important to highlight these works and clearly state steering identification difference for the community to understand the usefulness and differences of such steering. Please add discussion to compare your work with these steerable calibration methods, particularly highlight the differences and/or similarities in calculating the steering vectors.

2. The previously methods of reasoning trace steering are also one shot intervention, hence, the authors are requested to tone down on that part of the claim.

3. Need of a teacher model to extract reasoning traces is a weakness, as it requires additional distillation memory of a bigger model loading, as well as dependence on the choice of teacher model.

4. The extraction policy is very similar to activation steering as also noted by the authors, affecting the novelty of their key contribution.

5. Advance requirement to know the desired and non desired behavior is a weakness, as it is not clear on the guideline of knowing this. Also, identification of such things asks for human intervention that can be compared with that of the labeling in SFT, thus downgrading one of the benefits of cache steering.

[1] SEAL: Steerable Reasoning Calibration of Large Language Models for Free, COLM 2025.

[2] Activation Steering for Chain-of-Thought Compression, Arxiv 2025.

**Questions:**

1. Please add comparison of steering vector computation w.r.t [1, 2] and discuss differences in the steering process.

2. Please demonstrate the reasoning capability on other reasoning datasets as well.

3. Please compare the reasoning performance of an instruction tuned LLM with that of a rasoning training reasoning model.

[1] SEAL: Steerable Reasoning Calibration of Large Language Models for Free, COLM 2025.

[2] Activation Steering for Chain-of-Thought Compression, Arxiv 2025.

---

> ### Author Response · Authors · 2025-11-21
>
> Dear Reviewer,
>
> Thank you for highlighting the advantages of our approach, including style-level control without model retraining and the efficiency of a one-shot KV intervention compared to continuous activation edits. We respond to the your key concerns below.
>
> ---
>
> > “While this work of leveraging steering vector to induce reasoning thoughts in a model is different, there exists other steering method that are applied to reduce the reasoning thoughts (example: [1, 2]). Thus it is important to highlight these works and clearly state steering identification difference for the community to understand the usefulness and differences of such steering. Please add discussion to compare your work with these steerable calibration methods, particularly highlight the differences and/or similarities in calculating the steering vectors.”
>
> We thank the reviewer for highlighting these related papers – we will cite them in the revised manuscript.
>
> The papers pointed out by the reviewer, SEAL and Activation Steering for CoT Compression, both operate by continuously editing activations during decoding to reduce or calibrate reasoning length, typically in models that already produce CoT by default. Our work differs in several important ways:
> - **Target of intervention**. We intervene on the KV cache rather than the residual stream. This allows a single post-prefill modification applied to all layers, instead of repeated edits at a chosen layer.
> - **Goal of the method**. Our focus is on inducing CoT behavior in models that were not trained to reason, and on enabling controllable style transfer (stepwise, analogical, etc.). In contrast, SEAL/CoT-compression is applied to “thinking” models and aims to suppress or calibrate their excessive reasoning.
> - **Stability and sensitivity**. Activation-based methods require selecting a specific layer and are sensitive to coefficient tuning. We show that cache steering is robust to local variations in steering strength and does not require layer search.
> - **Vector computation**. The high-level extraction procedure (contrastive averages) is similar, but because the objectives differ, the data used to form the contrastive pairs differs as well, leading to vectors that encode entirely different behavioral signals.
>
> > “The previously methods of reasoning trace steering are also one shot intervention, hence, the authors are requested to tone down on that part of the claim.”
>
> We believe there might be a misunderstanding. When we claim that cache steering is a “one shot intervention”, we mean that it does not need to be applied continuously at every decoding step, which is also highlighted by the reviewer as the *strength* of the paper.
>
> We also note that both works cited by the reviewer apply their activation edits **continuously**, not as a single-step modification. Our intended distinction is that **cache steering performs one modification to fixed KV entries immediately after prefilling** (which can be precomputed and stored), after which decoding proceeds normally. Empirically, this single edit is stable and does not require layer search or repeated interventions. We intend to highlight that cache steering attains strong steering effects **without** introducing instability.
>
> If there are specific sentences the reviewer would like us to adjust or soften, we would be happy to update them directly.
>
> > “Need of a teacher model to extract reasoning traces is a weakness, as it requires additional distillation memory of a bigger model loading, as well as dependence on the choice of teacher model.”
>
> While our experiments used strong teacher models to produce the CoT data, **cache steering itself is fundamentally independent of any teacher models**. Steering vectors can be extracted from any source of contrasting reasoning examples (e.g., human-generated reasoning traces, already existing reasoning datasets, or smaller fine-tuned models). In other words, **the teacher model was used only for data generation**.
>
> To demonstrate that cache steering is not dependent on stronger models, we evaluated a 70B model on a subset of MATH dataset (Table 3). In this experiment, **we do not use any teacher**. Instead, we use the reasoning traces from the dataset itself. We observe a +7.41% improvement in accuracy over the baseline when applying cache steering **without using a teacher model** to generate the CoT traces.

---

> ### Author Response · Authors · 2025-11-21
>
> > “The extraction policy is very similar to activation steering as also noted by the authors, affecting the novelty of their key contribution.”
>
> The novelty of our contribution lies not in how the vector is computed, but in **when and where the intervention is applied**, and in the **practical advantages that follow from this shift**.
>
> Specifically, activation steering modifies the activations during decoding, requiring runtime edits to the residual stream, careful layer selection, and *typically* continuous intervention to remain effective. In contrast, cache steering performs a single post-prefill modification of the KV cache, after which decoding proceeds unchanged.
>
> This distinction is practically crucial:
> - No modification of the inference engine – compatible with any API or deployment setting that exposes KV caches.
> - Single, stable intervention instead of continuous activation edits.
> - All layers are steered simultaneously without layer search.
> - Near-zero overhead, as the steering vector can be precomputed and cached with no changes to the model architecture.
>
> Thus, while the extraction policy is similar (as with many steering methods), the mechanism and location of intervention are different, leading to substantially different stability, applicability, and deployment characteristics. Our main contribution is that this steering method is **practical enough to be actually used in production systems**.
>
> > “Advance requirement to know the desired and non desired behavior is a weakness, as it is not clear on the guideline of knowing this. Also, identification of such things asks for human intervention that can be compared with that of the labeling in SFT, thus downgrading one of the benefits of cache steering.”
>
> We appreciate the reviewer’s concern but would like to clarify that cache steering does **not** require detailed human labeling or fine-grained specification of behaviors. The method only needs **contrastive examples**, which can be as simple as presence vs. absence of a behavior (e.g., outputs with and without reasoning). This requirement is identical to that of prior work on activation steering and specifically the works referenced by the reviewer.
>
> Importantly, these contrastive traces are easy to obtain: they can come from (i) existing datasets, (ii) outputs of a stronger model, or (iii) a small number of human-written examples. In all our experiments, no manual labeling was required.
>
> Compared to SFT or RLHF, which require large-scale human annotation of thousands of examples, cache steering operates on a few dozen contrastive pairs and does not require any training or gradient updates. For this reason, we view the requirement as minimal and consistent with prior steering methods, rather than as a weakness.
>
> > “Please demonstrate the reasoning capability on other reasoning datasets as well.”
>
> We believe the current evaluation is already broad: we test on four standard benchmarks spanning arithmetic, scientific, commonsense, and physical reasoning (GSM8K, ARC-c, CSQA, PIQA) across six models, and further evaluate a larger model on two harder datasets (GPQA Diamond, MATH). These cover distinct reasoning skills, and the steering effects are consistent across them. We believe that additional datasets would not change the empirical conclusions.
>
> > “Please compare the reasoning performance of an instruction tuned LLM with that of a rasoning training reasoning model.”
>
> We are not entirely sure what specific comparison the reviewer intends here (e.g., SFT-trained CoT models, RL-trained long-CoT models, or R1-style distillations). Our goal in this paper is to show that cache steering reliably induces or modifies reasoning behavior in instruction-tuned models that were not explicitly trained to reason.
>
> We already evaluate cache steering on a wide range of instruction-tuned models (1B–70B), demonstrating that the method works consistently. Introducing a comparison with a separate “reasoning-trained” model family would answer a different research question and would not affect the validity of our contribution, which focuses on inference-time control, not on comparing training paradigms. If the reviewer had something else in mind, we would appreciate a clarification.
>
> ---
>
> We hope we have clarified all the concerns; if there are no remaining questions, we would kindly ask the reviewer to update the score.

---

### Official Review · Reviewer_XM7K · 2025-11-01

**Soundness:** 2
**Presentation:** 3
**Contribution:** 2
**Rating:** 4
**Confidence:** 4

**Summary:**

The paper proposed Cache Steering, which is a lightweight and easy-to-implement approach to steer language models for reasoning tasks. The approach focus on modification of the KV cache during inference with pre-computed steering vectors that calculated from contrastive set of prompt pairs. The shows advantages compared with activation steering in term of both accuracy and latency. The approach is also able to distill certain reasoning styles from teacher models.

**Strengths:**

1. The proposed Cache Steering is simple and easy-to-implement compared with other steering approaches. With little impact on latency, it shows better potential for real applications.
2. The experiments are comprehensive, covering a wide range of LLMs and various reasoning datasets.

**Weaknesses:**

1. Limited Performance: The proposed approach only achieved average accuracy improvement of less than 1% in Table 1, compared with CoT Prompt. Considering its performance vibration larger than 1% with different hyperparameters in figure 2, I'm not convinced that the proposed approach can further improve reasoning based on the commonly used Zero-shot CoT Prompt.
2. Lack of interpretability analysis: No theoretical or empirical analysis for explaining why and how the proposed approach works, leaving limited insights for further investigation.
3. Long CoT Models: Considering the promising reasoning performance of long CoT models like OpenAI o1 and DeepSeek-R1. I believe the value of the paper can be further improved if the proposed approach tested and discussed on long CoT models.
4. Typo issues: Equation in section 3.2 is not clear from my view. I suppose f_l(p^+) and f_l(p^-) are the key or value vectors obtained within the attention block of Transformer layer l. But the notations do not show this, and even the same notations are used for key and value vectors. The "which" is misspelled as "whcih" in line 358 and 694.

**Questions:**

Please see the weaknesses.

---

> ### Author Response · Authors · 2025-11-21
>
> Dear Reviewer,
>
> Thank you for the thoughtful feedback and for highlighting both the practical potential of cache steering, as well as the breadth of our experimental evaluation. Below, we address the concerns raised and provide additional clarification.
>
> ---
>
> > “The proposed approach only achieved average accuracy improvement of less than 1% in Table 1, compared with CoT Prompt.”
>
> Our goal is not to surpass zero-shot CoT prompting but to have it as a point of reference, which is widely used for lightweight interventions. The key result is that **cache steering matches CoT-level performance without using any CoT prompt, training, or added latency**, showing that the steering vector effectively induces the desired reasoning behavior.
>
> Although the average gain is small, in several settings, cache steering does produce substantial improvements over CoT (e.g., +1.4-4.9 on ARC-c, +4.6 on GPQA Diamond, +7.4 on MATH), which we view as a bonus rather than the primary objective. The method’s main advantage is enabling **prompt-free, training-free** control over reasoning behavior with minimal overhead. We would also like to highlight that the application of cache steering is not limited to inducing CoT, which is demonstrated by our style control experiments.
>
> > “No theoretical or empirical analysis for explaining why and how the proposed approach works, leaving limited insights for further investigation.”
>
> We would like to clarify that this paper is intended as an **empirical contribution**, similar to prior work on activation steering and cache interventions, which likewise provide little or no mechanistic interpretability analysis. Our goal here is to demonstrate that manipulating KV-cache directions is an effective and stable way to control model behavior.
>
> The method is motivated directly by prior findings on (i) contrastive activation directions capturing behavioral features, and (ii) KV-cache interventions affecting downstream generation. We build on this established intuition and provide extensive empirical evidence that the extracted KV vectors encode reasoning-related structure.
>
> However, for the revised version we follow the approach adopted by multiple previous studies on activation steering and add two-dimensional t-SNE projections of the positive and negative value representations for each layer in Appendix J.3, which indicate clear separation of the representations in almost all layers. This **clear separation in the latent space** of the model representations motivates the construction of the steering vectors.
>
> We fully agree that deeper mechanistic interpretability would be valuable future work, but we view it as outside the scope of the current empirical study. If the reviewer has specific analyses in mind, we would be very happy to include them in the revision.
>
> > “[...] I believe the value of the paper can be further improved if the proposed approach tested and discussed on long CoT models.”
>
> We agree that long-CoT models are highly capable reasoners, but this makes the **CoT-induction task fundamentally inappropriate for them**. These models are explicitly trained to always produce long, structured reasoning traces, so inducing CoT behavior via steering would not be measurable. One of our key goals is to show that cache steering can elicit reasoning behavior in models that do not naturally exhibit it, precisely the regime where long-CoT models no longer provide a meaningful test.
>
> > “Typo issues”
>
> Thank you for spotting these issues. We have corrected the “whcih” typos and clarified Section 3.2 by explicitly denoting $f_l^k(\cdot)$ and $f_l^v(\cdot)$ explicitly.
>
> ---
>
> We hope we have clarified all the concerns; if there are no remaining questions, we would kindly ask the reviewer to update the score.

---

### Author Response · Authors · 2025-12-03

Given the recent changes in the review process after the OpenReview incident, we would like to leave a summary for the AC highlighting what this paper contributes and how we addressed the reviewers’ concerns.

## **Core contributions**

This paper proposes **cache steering**, a simple, training-free mechanism for controlling behavior in LLMs by applying a **single post-prefill modification to the KV cache**. The key contributions are:

- **CoT induction and general behavior control**. We show that the method reliably induces chain-of-thought behavior and enables controllable transfer of reasoning styles (stepwise, causal chain, analogical, etc.) in frozen models.
- **A practical and more stable alternative to activation steering**. Cache steering requires no model changes, no continuous interventions, and no custom decoding loop, making it directly compatible with standard inference APIs like vLLM, llama.cpp, Ollama and with precomputed KV caches. This makes it possible to actually use it in production systems. In addition, we show that cache steering is less sensitive to local hyperparameter variations.
- **Broad, consistent empirical effects**. Across six instruction-tuned models (360M-70B) and multiple reasoning datasets (GSM8K, ARC-c, CSQA, PIQA, GPQA, MATH), cache steering successfully induces reasoning behavior and often improves accuracy over the baselines, while adding virtually no latency.

## **Responses to reviewers’ questions and additional analyses**

In the rebuttal and the revised paper we directly address every major concern raised by the reviewers. Following the feedback, we added several analyses and clarifications. Below we summarize how each issue was resolved.

**Novelty vs. prior KV-based and dynamic steering methods**

Reviewers requested clearer differentiation from prior work. We clarified the distinction between our method and:
- KV “coprocessor”, which requires training auxiliary models and modifying the inference stack. Cache steering instead performs a training-free intervention to existing KV representations.
- Dynamic activation steering, which needs continuous edits, multiple forward passes on each decoding step, KL-guided adjustments, memory banks or per-chunk interpolations between multiple vectors. Cache steering avoids vertical propagation of the steering effect through the layers by design, enabling a stable one-shot intervention on all layers at once. This makes the method more lightweight and practical.

**Robustness, hyperparameter sensitivity and motivation for the approach**

Reviewers asked for evidence beyond the original ablations, as well as deeper insight into why the method works. We expanded empirical evidence showing stability even further:
- **Layer/location ablation (App. J.1)**: Applying steering to different layers changes accuracy by ≤2%, supporting low sensitivity compared to activation steering, when applied to different layers.
- **Persistence analysis (App. J.2)**: Steering effects remain stable over long generations.
- **t-SNE analysis (App. J.3)**: Value-cache vectors from positive vs. negative reasoning traces cluster distinctly across layers, supporting that contrastive KV directions encode meaningful behavioral signals.

**Teacher model dependence**

Reviewers expressed concern about reliance on larger teacher models. We clarified that:
- Teacher models were used only to generate contrastive examples.
- Cache steering is independent from specific teacher models; we demonstrate this on the Llama-3.1-70B-Instruct + MATH subset, achieving a +7.41% gain using only original dataset reasoning traces.

## **Why we believe the paper should be accepted**
With the additional analyses added in the revision, the paper now provides a stronger, more complete empirical foundation. The method fills an important gap between activation-steering research and real inference-time control: it offers a **stable, lightweight and production-compatible** mechanism for manipulating behavior in LLMs, which is not the case with other methods.

We respectfully ask the AC to consider the paper for acceptance as a practical, empirically grounded contribution to behavior steering and low-cost reasoning control in LLMs.

---

### Meta-Review · Area_Chair_QyAL · 2026-01-07

**Summary:**

Reviewers acknowledged the paper’s practical motivation and the simplicity of cache steering as a lightweight, training-free alternative to activation steering, with strengths in ease of deployment, low latency, and broad empirical coverage across models and benchmarks. However, several concerns informed the decision. Multiple reviewers questioned the novelty relative to prior KV-cache and activation-based steering methods, noting that the core extraction procedure is similar and that the conceptual distinction was initially unclear. Others raised doubts about the significance of the empirical gains, which were often modest and sometimes within the range of hyperparameter variation, as well as the lack of evaluation on harder or more contemporary reasoning benchmarks. There were also concerns about limited mechanistic or interpretability insight, reliance on teacher-generated reasoning traces, and insufficient evidence that the method is consistently more robust than existing dynamic steering approaches. While the rebuttal clarified distinctions, added analyses, and strengthened empirical support, at least one reviewer remained unconvinced, and the remaining concerns collectively weighed against a clear consensus for acceptance.

**Reviewer Concerns:**

Addressed concerns:
The rebuttal addressed several substantive issues raised by the reviewers. In particular, the authors clarified the conceptual distinction between cache steering and prior activation- or KV-based methods, emphasizing the one-shot, post-prefill intervention and its practical deployment advantages. They added new ablations on layer/location sensitivity, persistence over long generations, and additional analyses (e.g., t-SNE visualizations) to support claims of robustness and stability. The dependence on teacher models was also mitigated by demonstrating that steering vectors can be derived from existing dataset traces without stronger teachers, and presentation issues and notation ambiguities were corrected.

Outstanding concerns:
Some concerns remain unresolved. The overall empirical gains are still modest in many settings, and the paper does not include evaluations on more challenging or contemporary benchmarks (e.g., complex coding or competition-style tasks), which some reviewers viewed as necessary to demonstrate practical impact. Mechanistic understanding remains limited, with interpretability analyses still largely descriptive rather than explanatory. Additionally, at least one reviewer remained unconvinced about the distinctiveness and usefulness of the approach despite the clarifications, indicating that questions about novelty and real-world significance persist.

**Reviewer Scores:**

Reviewer XM7K:
This reviewer’s concerns focused on the modest performance gains, limited interpretability, and the relevance of the method given strong long-CoT models. The rebuttal clarified the paper’s scope, corrected presentation issues, and added supporting analyses, but did not fundamentally change the empirical conclusions. I therefore believe the reviewer might have slightly softened their stance but would likely keep a similar score, perhaps moving from a 4 to a borderline 4–5.

Reviewer Pttn:
Reviewer Pttn explicitly stated after reading the rebuttal that they were unconvinced and would keep their reject recommendation. Even with fuller participation in the discussion, their emphasis on the lack of stronger benchmarks and additional evidence suggests their score would remain unchanged at a reject.

Reviewer qbMb:
This reviewer raised detailed concerns about novelty, robustness, and over-steering. The rebuttal directly engaged with these points and added new ablations and analyses addressing hyperparameter sensitivity and persistence. While some novelty concerns might remain, I believe this reviewer would likely increase their score modestly (e.g., from 4 to around 5–6), reflecting increased confidence in the method’s robustness and practical contribution.

---

### Decision · Program_Chairs · 2026-01-26

Reject